# Mimicry of emergent traits amplifies coastal restoration success

Ralph J. M. Temmink [1,23✉], Marjolijn J. A. Christianen [1,2,23], Gregory S. Fivash[3,23], Christine Angelini[4], Christoffer Boström [5], Karin Didderen[6], Sabine M. Engel[7], Nicole Esteban [8], Jeffrey L. Gaeckle [9], Karine Gagnon [5], Laura L. Govers[1,10,11], Eduardo Infantes [12], Marieke M. van Katwijk [13], Silvija Kipson[14], Leon P. M. Lamers [1,15], Wouter Lengkeek[1,6], Brian R. Silliman[16], Brigitta I. van Tussenbroek [17], Richard K. F. Unsworth [18,19], Siti Maryam Yaakub[20], Tjeerd J. Bouma[3,10,21,22] & Tjisse van der Heide[1,10,11✉]

Restoration is becoming a vital tool to counteract coastal ecosystem degradation. Modifying transplant designs of habitat-forming organisms from dispersed to clumped can amplify coastal restoration yields as it generates self-facilitation from emergent traits, i.e. traits not expressed by individuals or small clones, but that emerge in clumped individuals or large clones. Here, we advance restoration science by mimicking key emergent traits that locally suppress physical stress using biodegradable establishment structures. Experiments across (sub)tropical and temperate seagrass and salt marsh systems demonstrate greatly enhanced yields when individuals are transplanted within structures mimicking emergent traits that suppress waves or sediment mobility. Specifically, belowground mimics of dense root mats most facilitate seagrasses via sediment stabilization, while mimics of aboveground plant structures most facilitate marsh grasses by reducing stem movement. Mimicking key emergent traits may allow upscaling of restoration in many ecosystems that depend on self-facilitation for persistence, by constraining biological material requirements and implementation costs.

[1] Aquatic Ecology and Environmental Biology, Institute for Water and Wetland Research, Radboud University, Heyendaalseweg 135, 6525 AJ Nijmegen, The Netherlands. [2] Wageningen University & Research, Aquatic Ecology and Water Quality Management Group, P.O. Box 47, 6700 AA Wageningen, The Netherlands. [3] Department of Estuarine and Delta Systems, Royal Netherlands Institute for Sea Research and Utrecht University, 4401 NT Yerseke, The Netherlands. [4] Department of Environmental Engineering Sciences, Engineering School for Sustainable Infrastructure and Environment, University of Florida, PO Box 116580, Gainesville, FL 32611, USA. [5] Environmental and Marine Biology, Åbo Akademi University, Tykistökatu 6, 20520 Turku, Finland. [6] Bureau Waardenburg, Varkensmarkt 9, 4101 CK, 4100 AJ Culemborg, The Netherlands. [7] STINAPA, Barcadera 10, Bonaire, The Netherlands. [8] Bioscience Department, Swansea University, Singleton Park, Swansea, Wales SA2 8PP, UK. [9] Washington State Department of Natural Resources, Olympia, WA 98504, USA. [10] Conservation Ecology Group, Groningen Institute for Evolutionary Life Sciences, University of Groningen, 9700 CC Groningen, The Netherlands. [11] Department Coastal Systems, Royal Netherlands Institute for Sea Research and Utrecht University, 1790 AB Den Burg, The Netherlands. [12] Department of Marine Sciences, University of Gothenburg, Kristineberg Marine Research Station, Kristineberg 566, 45178 Fiskebäckskil, Sweden. [13] Department of Environmental Science, Institute for Water and Wetland Research, Radboud University, Heyendaalseweg 135, 6525 AJ Nijmegen, The Netherlands. [14] Department of Biology, Faculty of Science, University of Zagreb, Rooseveltov trg 6, 10000 Zagreb, Croatia. [15] B-WARE Research Centre, Toernooiveld 1, 6525 ED Nijmegen, The Netherlands. [16] Division of Marine Science and Conservation, Nicholas School of the Environment, Duke University, 135 Duke Marine Lab Road, Beaufort, NC, USA. [17] Reef Systems unit, Instituto de Ciencias del Mar y Limnología, Universidad Nacional Autónoma de México, 77580 Puerto Morelos, Quintana Roo, Mexico. [18] Project Seagrass, 33 Park Place, Cardiff CF10 3BA, UK. [19] Seagrass Ecosystem Research Group, College of Science, Swansea University, Swansea SA2 8PP, UK. [20] Department Ecological Habitats and Processes, DHI Water & Environment, 2 Venture Drive, 18-18 Vision Exchange, Singapore 608526, Singapore. [21] Building with Nature group, HZ University of Applied Sciences, Postbus 364, 4380 AJ Vlissingen, The Netherlands. [22] Faculty of Geosciences, Department of Physical Geography, Utrecht University, 3508 TC Utrecht, The Netherlands. [23] These authors contributed equally: Ralph J.M. Temmink, Marjolijn J.A. Christianen, Gregory S. Fivash. ✉email: r.temmink@science.ru.nl; tjisse.van.der.heide@nioz.nl

The decline and degradation of coastal ecosystems threatens biodiversity and the services that humans derive from these systems, such as carbon sequestration, coastal protection, pollution filtration, and the provisioning of food and raw materials[1,2]. Although government and nongovernmental stakeholders have invested hundreds of millions of dollars to protect threatened coastal ecosystems, their decline continues[3,4] due to the combined impacts of anthropogenic disturbances, including climate change-induced heat waves and increased cyclone intensity, as well as the direct impact from eutrophication, and coastal development[5–8]. As a consequence, salt marshes (42%), mangroves (35%), oyster reefs (85%), coral reefs (19%), and seagrass meadows (29%) have all declined globally in extent[4,9–12]. Conservation practitioners and policy makers are therefore searching for strategies to counter the mounting losses of coastal ecosystems and their vital services. Recent emphasis has focused on habitat restoration as a conservation intervention that could help answer this call[13,14]. However, coastal restoration requires innovation to increase its effectiveness, as current efforts to rebuild coastal wetlands and reefs are prone to failure and are often too expensive to be included as central features in large-scale conservation planning[15].

A recent, key innovation in coastal restoration revealed that harnessing self-facilitation between transplants can increase restoration yields[16]. Whereas earlier work showed that increasing planting density can increase restoration success[17,18], Silliman et al.[16] demonstrated that yields can be doubled simply by planting in clumps rather than applying commonly used plantation-style dispersed designs, while keeping overall density unchanged. Although this simple clumping technique has the potential to fundamentally change coastal restoration[12,19,20], facilitation-harnessing approaches could become particularly effective if the organism traits generating self-facilitation can be mimicked and, thus, produced and distributed at large scales. Such innovation would eliminate the need for acquiring large numbers of transplants that may harm donor populations or require expensive nurseries.

Each individual organism possesses traits, such as body size, or metabolic rates that play a large role in determining its fundamental niche[21–25]. However, when individuals spatially organize at the population level, this process may produce emergent traits that are defined as traits not expressed by any single individual or small clone, but only emerge at the organizational level of the group or a large clone[26]. For example, individual mussels and oysters aggregate into reefs that ameliorate wave stress and reduce predation[27], while expansive seagrass and cordgrass clones form contiguous meadows that decrease erosive and anoxic stress[16,28,29]—properties that cannot be generated by small clones or individuals in isolation. A consequence of reducing physical stressors through emergent traits is that the realized niche may exceed the fundamental niche defined by the individual traits, allowing an established population to inhabit conditions otherwise unsuitable for a single individual or a small clone[21]. However, to enable establishment under such conditions, a critical threshold for population size and/or density thus needs to be overcome[29]. Under natural conditions, establishment may occur during a Window of Opportunity—a sufficiently long period of exceptionally calm conditions during which isolated individuals or small clones can settle and grow[30]. However, such Windows are relatively rare and, as a consequence, natural reestablishment processes often take decades or longer. In such systems, restoration can act to accelerate this temporal delay by transplanting sufficiently large populations or clones[16]. However, transplantation at the required scale is often infeasible because of the resources and time required to harvest or cultivate, and then transplant sufficient material.

Here, we propose to address this limitation and investigate a restoration concept, inspired by recent advancements in transplant designs[16] and based on engineering, in which we mimic key emergent traits that generate self-facilitation. We developed biodegradable establishment structures with the aim to enhance the survival and growth of small salt marsh grass and seagrass transplants (Fig. 1 and Supplementary Fig. 1), thereby minimizing costs and the need for often limited donor material. These complex 3D-structures ameliorate hydrodynamic energy from waves and flow, and stabilize and accumulate sediment, thereby mimicking critical emergent traits—i.e., dense aggregations of roots or stems—that invoke self-facilitation naturally generated by established conspecifics, such as observed in sufficiently large and dense salt marsh and seagrass patches. Earlier observational and experimental work revealed that root mats of both seagrass and cordgrass are important for stabilizing sediment[16,28,29,31,32]. Attenuation of hydrodynamic energy and resulting sediment accumulation by aboveground stems, on the other hand, is much stronger in patches of stiff salt marsh cordgrass stems compared to drag avoiding, flexible seagrass shoots[33–35]. Therefore, we hypothesize that mimicry of belowground root mats of established vegetation patches should benefit both seagrass and cordgrass, while mimicking drag reduction due to attenuation of hydrodynamic energy by aboveground stems should be particularly beneficial for cordgrass. The structures should allow small transplants to survive and expand within the structure, and are designed to naturally degrade once the transplants are sufficiently established.

To investigate our concept, we apply the structures belowground to simulate sediment stabilization by vegetation root mats, and aboveground to reduce hydrodynamic energy in two seagrass ecosystems in temperate Sweden (*Zostera marina*) and tropical Bonaire (*Thalassia testudinum*), and in two cordgrass salt marsh systems in temperate Netherlands (*Spartina anglica*) and subtropical US Florida (*Spartina alterniflora*, Fig. 1). In addition, we combine field measurements on sediment stability with laboratory flume experiments on cordgrass stem movement to unravel the mechanisms underlying the results from our restoration experiments in the field. Our study shows that mimicking emergent traits that generate facilitation increases plant growth and survival, thereby enhancing restoration yields. This approach may allow upscaling of restoration in many ecosystems that depend on self-facilitation for persistence by limiting donor material and implementation costs.

## Results

**Experimental results**. Over periods of 12–22 months, above- and belowground structures positively affected survival and growth of seagrass and cordgrass transplants in both temperate and (sub) tropical regions (for site-specific details see Supplementary Table 1). In general, survival of both seagrass and cordgrass was low or zero in controls that lacked the establishment structure. Seagrass survival peaked when transplanted in belowground structures, while cordgrass transplant survival was highest when transplanted in aboveground structures (Fig. 2). For both seagrass sites, transplant survival was similar, with $100 \pm 0\%$ ($\pm$SE) in the belowground structures, $75 \pm 25\%$ in the aboveground structures, and only $20 \pm 20\%$ in the controls (without structures). Cordgrass survival was $100 \pm 0\%$ and $28 \pm 18\%$ in the above- and belowground structures in the Netherlands, respectively, while survival was $75 \pm 16\%$ in both above- and belowground structures in Florida. In controls, cordgrass transplant survival was zero at both sites.

Above- and belowground structures also positively affected shoot number and the maximum lateral expansion of seagrass

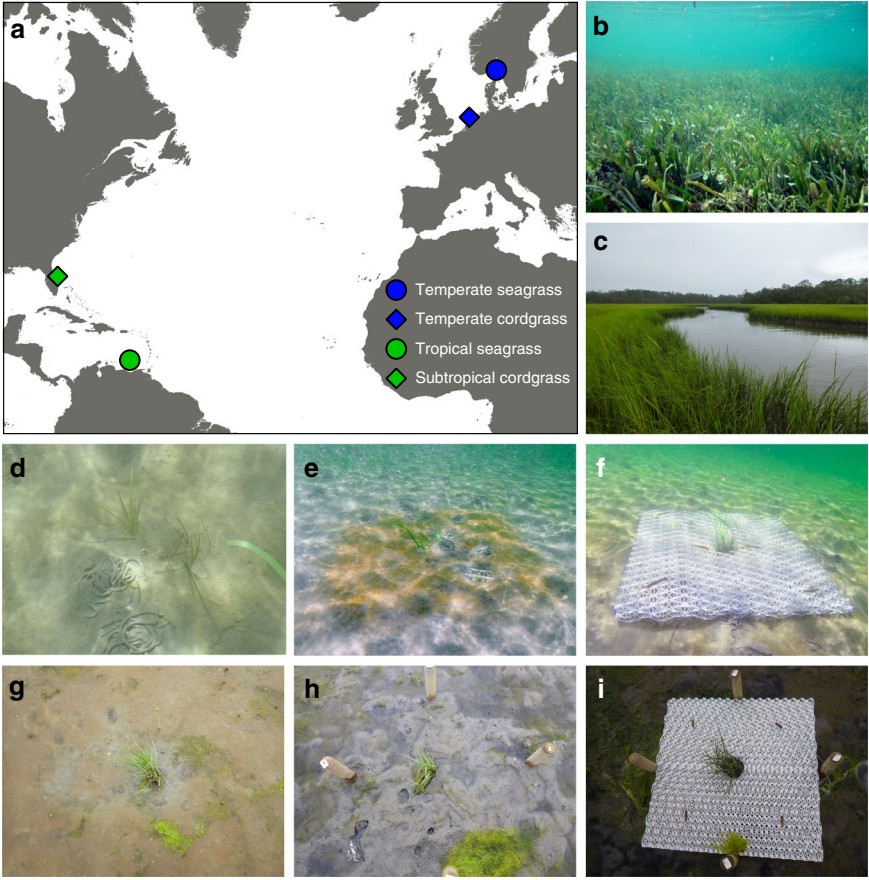

**Fig. 1 Field sites and experimental setup. a** The locations of the field sites. Blue circle: temperate *Zostera marina* (Sweden), green circle: tropical *Thalassia testudinum* (Bonaire), blue diamond: temperate *Spartina anglica* (the Netherlands), and green diamond: subtropical *Spartina alterniflora* (Florida, USA).
**b**, **c** Mature seagrass and salt marsh ecosystems; **d–f** bare, belowground, and aboveground establishment structures with seagrass transplants in Sweden after setup; **g–i** the same setup with cordgrass transplants in the Dutch salt marsh. Map data made with Natural Earth by RJMT.

and cordgrass in both temperate and (sub)tropical regions. Seagrass benefited most from belowground structures, whereas cordgrass was most strongly facilitated by aboveground structures (Figs. 3 and 4). Seagrass shoot numbers were highest in belowground structures with 30.1 ± 5 shoots for *Z. marina* in Sweden and 15.5 ± 2 shoots for the slower-growing climax species *T. testudinum* in Bonaire. Shoot counts in aboveground structures were 4.6 times (6.5 ± 3 shoots) and 2.2 times (6.8 ± 3 shoots) lower for Sweden and Bonaire, respectively, and controls had even lower shoot counts with 0.5 ± 0.5 and 0.25 ± 0 (Fig. 3a, b). By contrast, cordgrass transplants produced the most shoots in aboveground structures (47.5 ± 22 and 6.8 ± 2 shoots in the Netherlands and Florida, respectively, Fig. 3c, d), while numbers in belowground structures were 53 times (0.9 ± 1 shoots) and 2.6 times lower (2.6 ± 0.8 shoots). As these shoot numbers are below the initial count in the transplants (17.6 ± 0.4 and 4.9 ± 0.2 shoots/transplant in the Netherlands and Florida, respectively), these results suggest that belowground structures do not sufficiently facilitate cordgrass to warrant long-term success. Finally, no shoots were presents at controls in the salt marsh sites.

Similar to the number of shoots, maximum lateral expansion was highest in belowground structures for seagrass, and highest in aboveground structures for cordgrass (Fig. 4a, d). In controls, maximum lateral expansion was on average <5 cm. For seagrass, maximum lateral expansion in Bonaire was 1.6 times higher in below- (57 ± 11 cm) compared to aboveground structures (36 ± 13 cm), while it was six times higher in below- vs. aboveground structures in Sweden (30 ± 7 cm and 5 ± 4 cm, respectively). Maximum lateral expansion by cordgrass reached 31.6 ± 9 and 42.6 ± 12 cm in aboveground structures in the Netherlands and Florida, respectively, which was 2.5 and 2.1 times higher than belowground structures. Cordgrass expansion was zero in all controls, because the transplants did not survive.

Additional measurements on sediment stability in the seagrass field experiments and cordgrass stem movement in laboratory flume experiments exposed the mechanisms underlying the observed differential responses of seagrass and cordgrass to the mimicry treatments. Field measurements using sediment-burial pins in both Sweden and Bonaire seagrass beds demonstrated that sediment movement was highest in controls, and was reduced on average by 37% ± 18 in the aboveground establishment structures (Fig. 5e). The belowground structures, however, proved much more effective, as they reduced sediment movement by 77% ± 22 and 63% ± 21 compared to controls and aboveground structures, respectively.

While clearly underperforming compared to belowground structures with regard to sediment stabilization, aboveground structures were highly effective in mitigating wave-imposed movement of stiff cordgrass stems when subjected to waves in flume experiments (Fig. 5f). Specifically, stem movement was reduced 1.3 times by the aboveground structure compared to controls under low wave energy (significant wave height, $H_{1/3} = 25$ mm), and this mitigating effect increased to 1.4 times under medium wave energy conditions ($H_{1/3} = 50$ mm), and 1.8-times under high-wave energy conditions ($H_{1/3}$: 70 mm; Supplementary Fig. 2).

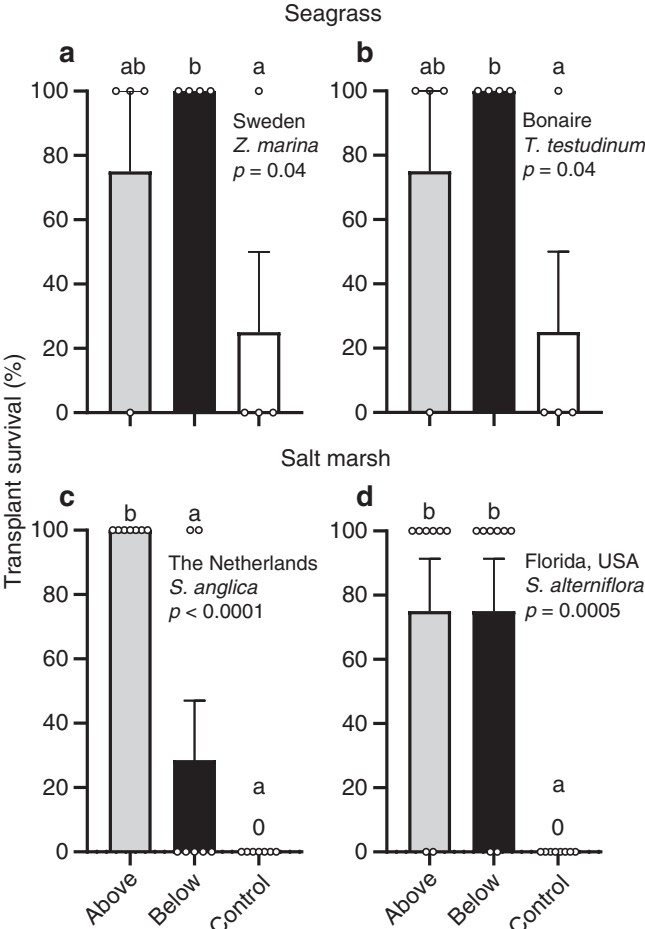

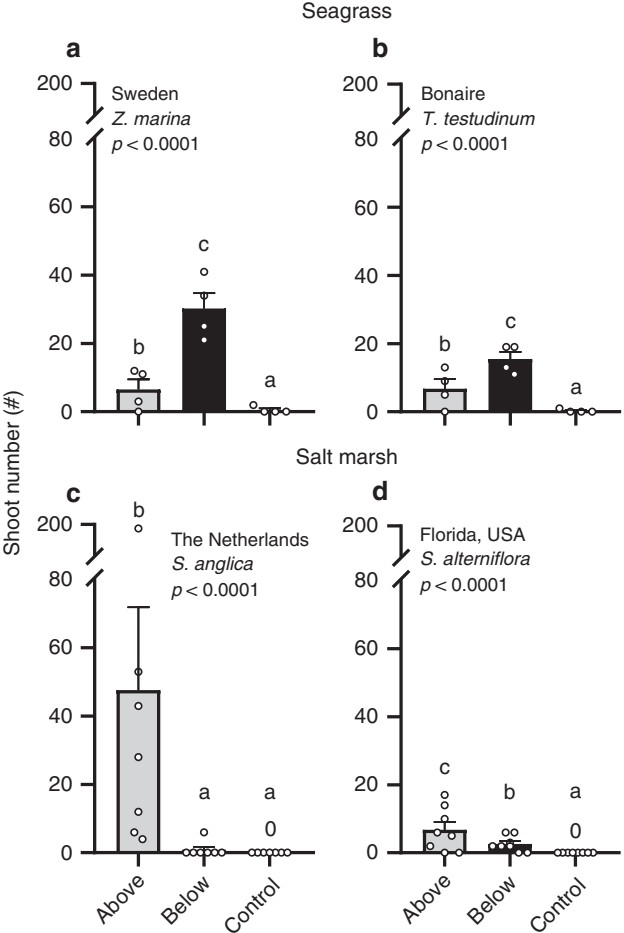

**Fig. 2 Transplant survival. a, b** Seagrass transplant survival in Sweden ($n = 4$) and Bonaire ($n = 4$) in above- (gray) and belowground structures (black), and controls (white). **c, d** Cordgrass transplant survival in the Netherlands ($n = 7$) and Florida ($n = 8$). Note that survival at both seagrass sites was identical. Data are presented as mean values + SEM. Exact $p$ values are shown for treatment effects when $p > 0.0001$ (two sided). Significant contrasts are indicated by different letters ($p < 0.05$, Benjamini–Hochberg corrections for multiple comparisons). Results of the statistical analyses are presented in Supplementary Table 3. Source data are provided as a Source Data file.

**Fig. 3 Seagrass and cordgrass transplant shoot numbers. a, b** Seagrass shoot counts in Sweden ($n = 4$) and Bonaire ($n = 4$) in above- (gray) and belowground structures (black), and controls (white). **c, d** Cordgrass shoot counts in the Netherlands ($n = 7$) and Florida ($n = 8$). Data are presented as mean values + SEM. Exact $p$ values are shown for treatment effects when $p > 0.0001$ (two sided). Significant contrasts are indicated by different letters ($p < 0.05$, Tukey corrections for multiple comparisons). Results of the statistical analyses are presented in Supplementary Table 3. Source data are provided as a Source Data file.

**Cost feasibility.** To illustrate the potential scalability of trait-based mimicry as a general approach, we calculated construction costs for four scenarios per ecosystem in which we upscale our specific technique as an example. The costs to restore vegetated coastal ecosystems range from 5000 to 280,000 US$/ha (Supplementary Table 2), depending on the plant expansion rate and the restoration period (5 or 10 years). For instance, using fast-growing species and a long restoration period results in lowest costs with 6250 and 5000 US$/ha for salt marsh and seagrass systems, respectively (Supplementary Table 2). Costs increase four times to 25,000 and 20,000 US$/ha when shortening the restoration period to 5 years. Selecting slow-growing species and using a short restoration period result in the highest costs of 100,000 and 280,000 US$/ha for salt marsh and seagrass systems, respectively (Supplementary Table 2).

## Discussion

Organisms living in harsh environments, such as coastal zones, have been found to often reduce physical stress through emergent traits that broaden the realized niche of individuals to exceed their

fundamental niche, allowing them to inhabit otherwise unsuitable conditions[21]. Here, we demonstrate that mimicking key emergent traits successfully simulates this positive density-dependent facilitation, thereby increasing growth and survival of isolated transplants and enhancing restoration yields. At present, erosion is an increasing problem along coastlines in general, and at degraded sites that require restoration in particular[36]. To combat this pervasive challenge, hard structures from shells or concrete are often applied to provide stable substrates necessary to stimulate reef formation[37–43], while sediment stabilization measures have been used to support vegetation establishment[44,45]. Our approach builds upon these efforts by experimentally demonstrating that tailor-made mimicry of species-specific key emergent traits—identified from past ecological studies—facilitates the establishment of different habitat-forming species. Specifically, our results highlight that by mimicking dense cordgrass patches that attenuate hydrodynamic energy[35,46] or extensive seagrass root mats that improve sediment stability[31], restoration success can be greatly enhanced and, in many cases, may turn failures into successes.

Our experimental results demonstrate that by mimicking mature roots mats or dense patches of stiff plant stems, survival

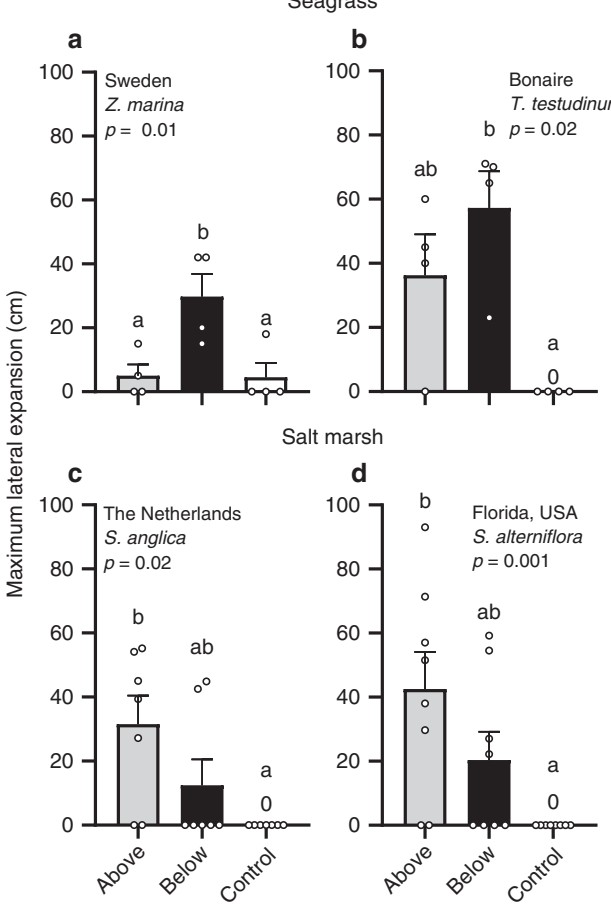

**Fig. 4 Maximum lateral expansion of the transplants. a, b** Seagrass expansion in Sweden ($n = 4$) and Bonaire ($n = 4$) in above- (gray) and belowground structures (black), and controls (white). **c, d** Cordgrass expansion in the Netherlands ($n = 7$) and Florida ($n = 8$). Data are presented as mean values + SEM. Exact $p$ values are shown for treatment effects when $p > 0.0001$ (two sided). Significant contrasts are indicated by different letters ($p < 0.05$, Benjamini–Hochberg corrections for multiple comparisons). Results of the statistical analyses are presented in Supplementary Table 3. Source data are provided as a Source Data file.

and expansion of otherwise vulnerable transplants were much higher. By simulating root mats using belowground establishment structures, sediments were stabilized similar to what is observed in natural matured patches[47,48]. This, in turn, enhanced both cordgrass and seagrass survival, as well as seagrass growth. Furthermore, cordgrass restoration yields were enhanced more by aboveground relative to belowground establishment structures, while in seagrass trials we found opposite results. Our additional mechanistic experiments demonstrate that, in mimicking established dense stands of stiff cordgrass stems, aboveground establishment structures reduced movement of small cordgrass transplants, similar to the movement reduction experienced by salt marsh grasses in natural, mature patches[46,49]. Moreover, this facilitating effect became increasingly apparent with rising wave heights, emphasizing the increasing importance of positive interactions under high physical stress, such as at our field sites where wave heights during extreme conditions exceed those simulated in our flume (field: 0.08–0.57 m; flume: 0.03–0.07 m; Supplementary Table 1; Supplementary Fig. 2). Strikingly, in contrast to the stiff cordgrass stems, seagrasses benefitted much less from aboveground stabilization. Most likely this is because flexible seagrass shoots typically move with the flow rather than resist it[34,35], a trait

that may have been hampered by the aboveground structures, as they limit shoot movement and hence the ability of seagrass stems to avoid drag (Fig. 5). In addition, the increase in shoot number differed considerably depending on whether a faster- (e.g., *Z. marina*) or slower-growing (e.g., *T. testudinum*) species was introduced. Combined with the finding that belowground structures provide better sediment stabilization compared to aboveground treatments, these differences in stem traits explain the differential, ecosystem-specific results, highlighting the need to tailor emergent trait-based restoration approaches to specific habitat-forming species and environmental conditions.

Recent experimental work from Dutch and US salt marshes demonstrates that harnessing beneficial species interactions through design can double restoration yields, because self-facilitation is instantaneously created by clumping transplants[16]. Although clumping into larger patches can enhance transplant survival, it diminishes the transplants' potential to expand laterally, because the relative edge length along which the vegetation can expand decreases isometrically with increasing patch size[50]. Therefore, clumped configurations require more transplant units to achieve lateral outgrowth rates that sufficiently warrant recolonization. Here, we show that by deploying transplants inside establishment structures, our salt marsh transplant size was nine times smaller compared to the earlier applied clumped transplant design[16], greatly reducing the need for donor material and avoiding potential damage to donor sites or demands on nurseries to cultivate transplants. As clumping has also been previously found to benefit seagrass transplants[51], and a review and separate global analysis showed that small-scale facilitations and large-scale approaches will generally benefit seagrass restoration success[17,52], our finding suggests that the use of establishment structures may be more beneficial for seagrass restoration.

Although restoration is increasingly advocated to serve as an important strategy to halt and reverse coastal ecosystem losses worldwide, current high costs and unpredictable outcomes make it a risky investment, hampering large-scale application. For example, the costs of restoring terrestrial ecosystems such as grasslands, woodlands, temperate, and tropical forests range from 500 to 5000 US$/ha[53], on average, at spatial scales ranging from <1000 to >100,000 ha[54]. By contrast, restoration of coastal ecosystems typically occurs at spatial scales of 0.1–1000 ha with costs ranging from 15,000 to 1,000,000 US$/ha for vegetated coastal ecosystems, and with coral reef restoration typically being even more expensive (up to 5,500,000 US$/ha)[15]. Our results highlight that under harsh conditions where self-facilitation is important, mimicry of self-facilitating, emergent traits can increase both restoration success, and cost-effectiveness, particularly when using fast-growing species and accepting a long restoration period (Supplementary Table 2). For instance, using patch-wise application of the mimics from this study to support establishment and lateral expansion of individual salt marsh or seagrass transplants would cost 5000–280,000 US$/ha, depending on the plants expansion rate and the period (5 or 10 years) within which restoration practitioners seek to achieve coalesced vegetation stands (Supplementary Table 2). This illustrates that trait-based mimicry design may be particularly helpful in harsh conditions where restoration is inherently failure prone and expensive. By contrast, the approach is likely unsuitable for benign conditions, where seeding or dispersed transplant designs may prove to be more cost-efficient alternatives[16,17,55] or when the environmental conditions are too harsh to be sufficiently mitigated by emergent traits of an established population. In the latter case, only permanent protection measures, such hard defense structures, would provide a long-term feasible option to allow vegetation development. Finally, large-scale application should also be carefully judged in ecosystems that are suitable from an environmental

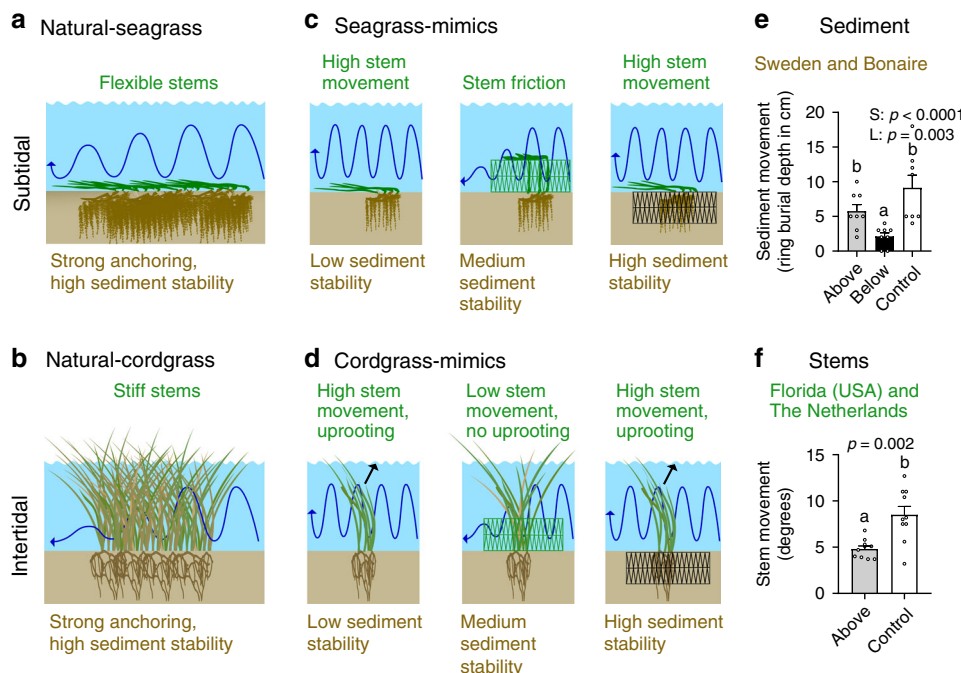

**Fig. 5 Species-specific facilitation mechanisms.** Both cordgrass and seagrass increase sediment stability with their root mats, but stiff cordgrass stems also attenuate hydrodynamic energy (blue arrow), while flexible seagrass shoots avoid drag by bending (**a**, **b**). Small cordgrass and seagrass transplants cannot self-facilitate, making them vulnerable to uprooting (black arrow). Application of trait-based mimicry allows simulating self-facilitation naturally occurring in mature vegetation stands (**c**, **d**). Belowground establishment structures simulate a dense root mat, while aboveground structures mimic dense patches of stiff cordgrass stems. Field measurements in Sweden and Bonaire confirm sediment stabilization by aboveground establishment structures, but even more by belowground structures (**e**). Flume experiments demonstrate that aboveground structures greatly reduce cordgrass stem movement when subjected to 70-mm-high waves ((**f**), $n = 10$). Panel **e** shows sediment mobility grouped for Sweden and Bonaire (ring burial depth in cm, $n = 8$). Main effects (S structure, L location) are shown with $p$ values (two sided); significant contrasts with letters ($p < 0.05$, Tukey corrections for multiple comparisons). Exact $p$ values are shown when $p > 0.0001$. Data are presented as mean values + SEM. Results of the statistical analyses are presented in Supplementary Table 3. Source data are provided as a Source Data file. Symbols for diagrams courtesy of the Integration and Application Network, IAN Image Library (ian.umces.edu/imagelibrary/).

perspective, but considered vulnerable regarding for instance water and sediment quality, or the intermediate-term fate of biodegradable material. In such cases, permitting and mitigation measures could result in a prolonged project duration and higher costs.

While our experimental results show that the establishment structures used here can enhance restoration success, and costs are such that upscaling is feasible, our mimicry of emergent traits is still relatively crude, highlighting a potential need for optimization. 3D-printing may, for example, prove a very useful tool to develop biodegradable prototypes as it opens up virtually infinite design possibilities and allows for fine details at the microscale[56,57]. To enable such optimization, identifying the bottlenecks that hamper establishment of the target species should be the first step[19,52,58]. Next, it should be established whether the target species, or species that mutualistically interact with the target species[59], possesses emergent traits that mitigate these bottlenecks, after which the establishment structure's design can be improved to more accurately simulate these traits. In many cases, however, there may be multiple solutions to emulate a certain emergent trait, turning such a design optimization goal into a complex problem with many potential solutions, particularly when there are multiple traits to be considered. In engineering design, such a complex problem is often approached using a morphological analysis that allows exploration of all possible solutions for the combinations of functions one aims to achieve[60]. For restoration, morphological analysis may help design structures that simultaneously ameliorate multiple emergent trait-mitigated bottlenecks, such as wave attenuation

combined with sediment stabilization by coastal vegetation[38,61], or provisioning of attachment substrate combined with predation shelter by oysters and mussels[28,38,62–67].

Apart from marshes and seagrass meadows, many marine, freshwater, and terrestrial ecosystems, including coral and shellfish reefs, mangroves, rivers, peatlands, and (semi-)arid lands, are dominated by species that self-facilitate and whose colonization success often depends critically on overcoming establishment thresholds[27,68,69]. Consequently, restoration of such ecosystems faces issues similar to those in salt marshes and seagrass meadows. For example, restoration of mangroves via seeds in dynamic environments with unstable sediments may profit from the use of temporary mimicry of established mangrove trees[68]. Furthermore, restoration of shellfish and coral reefs has been found to be hampered by a lack of suitable settlement substrate, often combined with high predation pressure on recruits due to a lack of habitat complexity[42,70]. In such cases, structures that mimic attachment substrate provisioning and predation reduction benefits typically generated by established reefs (e.g., in texture and crevice size or scaring prey with predator cues) may be helpful[19,42,52,70,71]. Hence, we suggest that our trait-based approach may inspire follow-up research investigating how mimicry of emergent traits by habitat-forming species may enhance establishment and restoration yields in harsh environments.

## Methods
**Study sites**. Fieldwork was conducted at bare restoration sites between 2016 and 2019 (Fig. 1 and Supplementary Table 1), where vegetation was historically present.

Both salt marsh sites were intertidal, with an average flooding regime of twice a day, whereas both seagrass sites were permanently submerged. The Dutch salt marsh site and both seagrass sites were characterized by sandy sediments, while the Florida marsh site was characterized by a mix of silt and sand. All four sites were selected for their relatively exposed hydrodynamic conditions (Supplementary Table 1), and mobile sediments—conditions where self-facilitating traits of seagrass and salt marsh plants should be beneficial.

**Restoration experiment.** We randomly assigned one of three treatments to each plot in a randomized block design: aboveground establishment structure, belowground establishment structure or control ($n = 7$ replicate blocks for the Netherlands, $n = 8$ for Florida, $n = 4$ for Bonaire and Sweden). In each system, belowground establishment structures were buried, completely sub-surface, in the sediment to simulate dense seagrass or cordgrass root mats, while aboveground structures were placed on the sediment surface to simulate dense patches of stiff (i.e., cordgrass like) vegetation stems.

Plots were spaced >2 m apart in areas with bare sediment, where vegetation was previously mapped, but had disappeared. For each system, transplants were obtained from neighboring stands. Cordgrass transplants were collected as plugs[16] ($10 \times 15$ cm, diameter × height), and contained $17.6 \pm 0.4$ and $4.9 \pm 0.2$ shoots in the Netherlands and Florida, respectively. Each plug was manually transplanted level with the sediment surface in the center of each plot. A 10-cm circle in the middle was cut in the center of every establishment structure. Seagrass transplants were manually collected as rhizomes or ramets with apical growing tips. In each plot, three rhizomes were hand planted in the center with growing tips pointing outwards, resulting in $2.9 \pm 0.2$ shoots for Sweden and $7.7 \pm 0.3$ shoots for Bonaire at the start of the experiment. Rhizomes were anchored using u-shaped pins (20 cm length) tied to the rhizome with cable ties. The experiments ran between 12 and 22 months (Supplementary Table 1), after which transplant survival was monitored, while shoot number and the maximum lateral outgrowth were determined as proxies for growth. Lateral outgrowth was measured as the straight-line distance from the plot center to the newest shoot at the end of the longest rhizome.

Establishment structures consisted of BESE elements (https://www.bese-elements.com) composed of biodegradable potato-waste-derived Solanyl C1104M (Rodenburg Biopolymers, Oosterhout, the Netherlands). Single sheets ($91 \times 45.5 \times 2.0$ cm; 0.44 kg, surface:volume ratio 80 m$^2$/m$^3$) can be clicked together to form a modular complex 3D-structure (Supplementary Fig. 1). For the purpose of our study, three sheets were combined to form a 6-cm high 3D honeycomb-shaped matrix. Next, half a circle with a diameter of 10 cm was removed from the middle of the longest side of the sheet using a disk grinder. Combining two of such structures thus yielded a 6-cm high $91 \times 91$ cm establishment structure with a 10-cm circle in the middle (Supplementary Fig. 1).

In the field, each $91 \times 91 \times 6$-cm establishment structure, was either buried 6 cm into the sediment (treatment: belowground establishment structure, Fig. 1e, h) or placed on top of the sediment (treatment: aboveground establishment structure, Fig. 1f, i) to form a plot with a cordgrass plug or seagrass transplants in the center circle. In the Netherlands, establishment structures were secured using two 50-cm long L-shaped steel rebar anchors that were pushed through the structures into the sediment, combined with four 100-cm long chestnut poles (7 cm diameter) positioned along the four sides, cross-connected over the structures with plastic coated steel wire. In Florida, each establishment structure was secured using five 100-cm long L-shaped rebar anchors. In Bonaire and Sweden, each establishment structure was secured using six 90-cm long rebar anchors. Every control plot was marked with a bamboo stick or a rebar.

**Mechanistic measurements and experiments: sediment and stem movement.** Sediment movement was measured in the Bonaire and Sweden experiments by placing sediment-burial pins for a month in the center of each plot. Specifically, 50-cm long stainless pins were driven 40 cm into the ground[72]. Next, a flat ring was placed around the pin on the sediment surface, after which the distance between the upper tip of the pin and the sediment level was measured. Over the course of the following month, the ring moved downward each time the sediment became unstable. As a proxy of sediment mobility, we therefore measured the distance between the sediment level and the ring.

We used a wave flume to show the principle of how cordgrass stem movement was affected by the aboveground establishment structure. The flume, located at NIOZ (the Netherlands), is 17.5-m long, 0.6-m wide, and 0.4-m high water channel in which regular waves can be generated by a vertical wavemaker driven by a back-and-forth moving piston[34]. It has a 2-m long test section with a transparent side window, allowing direct observations and recording of stem movement. The test section has an adjustable bottom allowing a 0.3-m deep sediment bed, which we constructed from coarse sand. Behind the test section, waves are dampened by a porous gentle slope[73]. In the experiment, we used 30-PSU seawater from the Eastern Scheldt. Water height within the flume was maintained at 30 cm.

Within the test section, we placed 15 162-mm long cordgrass mimics, resembling natural cordgrass vegetation, fixed to a mesh[34,46] in the 10 cm diameter opening of the aboveground establishment structure (dimensions: $90 \times 60 \times 6$ cm (L × W × H)) or at a bare sediment control. Next, mimics were subjected to 25, 50, and 70-mm high waves, while stem movement was recorded from the side for 60 s by a video camera

(Garmin Virb Ultra 30) at 10 frames/s. For each run, the maximum angle of 10 random shoots were measured in 50 frames over 50 s using ImageJ[74].

**Statistical analyses.** Each field site was separately analyzed for treatment effects (i.e., control, above-, and belowground establishment structure) on transplant survival, maximum lateral expansion, and shoot number. Although the included seagrass and marsh species share important traits, each site harbors distinctly different species due to the differences in climate conditions. We therefore statistically analyzed each site separately. Transplant survival was analyzed using General Linear Models with a binomial distribution, followed by pairwise comparisons with Benjamini–Hochberg corrections of the significance level. Shoot numbers were analyzed with Generalized Linear Mixed Models with a Poisson distribution and block as random effect[75], followed by Tukey post hoc tests[76]. Poisson models were checked for overdispersion, and if unsatisfactory, a negative binomial model was used (Sweden data). Maximum lateral expansion was analyzed non-parametrically using Kruskal–Wallis tests followed by Dunn tests with Benjamini–Hochberg corrections of the significance level for multiple comparisons, as assumptions for normality could not be met. Sediment movement data (square root transformed) were analyzed using a Linear Mixed-Effect Model with treatment and location as factors, and block as a random effect, with treatment differences determined by a Tukey test. Stem movement measured in the flume experiment was analyzed using a t-test with unequal variances. Data were analyzed with R version 3.6.0[77].

**Cost-feasibility analysis.** To illustrate the potential applicability of trait-based mimicry, we calculated construction costs for a number of scenarios in which we upscale our specific technique as an example. Specifically, we considered the following four scenarios for both seagrass and salt marshes: (1) short recovery time, fast plant growth, (2) long recovery time, fast plant growth, (3) short recovery time, slow plant growth, and (4) long recovery time, fast plant growth. We chose these specific scenarios because they reflect the trade-off between construction costs, species selection, and restoration time that restoration practitioners may face when applying this method. Based on actual restoration projects[15], we chose two restoration periods in which complete recovery should be accomplished; i.e., 5 (short) vs. 10 (long) years to establish a continuous vegetation stand. In addition, we selected two contrasting lateral extension rates of transplants (i.e., fast vs. slow growth) to illustrate the effect of species selection on the costs. Construction costs are extrapolated from actual costs in our experiments. Lateral extension rates are based on data from this work, combined with additional data from literature[16–78,79,80] (Fig. 4 and Supplementary Table 2). In each scenario, the approximately1-m$^2$ establishment structures were assumed to be spread out evenly across space. Their required initial cover (% of a hectare) depends on the selected restoration period and expansion rate of plant species.

**Reporting summary.** Further information on research design is available in the Nature Research Reporting Summary linked to this article.

## Data availability

All data that support the main findings of this study are available via the Data Archiving and Networked Services (DANS) EASY (https://doi.org/10.17026/dans-xx2-s4c6)[81]. In addition, the source data of Figs. 2–5 and Supplementary Fig. 2 are provided as a Source Data file. All other relevant data are available upon request. Source data are provided with this paper.

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

## Acknowledgements

The authors thank all the volunteers for assistance in the field. R.J.M.T., G.S.F., K.D., and W.L. were funded by NWO/TTW-OTP grant 14424, in collaboration with private and public partners: Natuurmonumenten, STOWA, Rijkswaterstaat, Van Oord, Bureau Waardenburg, Enexio, and Rodenburg Biopolymers. M.J.A.C., S.K. and K.G. were funded by EU-H2020 project MERCES grant 689518. M.J.A.C. was funded by NWO-Veni grant 181002. T.H. was funded by NWO/TTW-Vidi grant 16588. B.R.S. was funded by a grant from the Lenfest Ocean Program and from Duke Restore. C.B. was funded by the Åbo Akademi University Foundation SR.

## Author contributions

R.J.M.T., M.J.A.C., G.S.F., C.A., K.D., W.L., T.J.B., and T.H. designed the experiments. R.J. M.T. and M.J.A.C. coordinated the fieldwork. R.J.M.T. and G.S.F. performed the flume experiment. All authors (R.J.M.T., M.J.A.C., G.S.F., C.A., C.B., K.D., S.M.E., N.E., J.L.G., K.G., L.L.G., E.I., M.M.K., S.K., L.P.M.L., W.L., B.R.S., B.I.T., R.K.F.U., S.M.Y., T.J.B., T.H.) were involved in carrying out the field experiments. R.J.M.T., B.R.S., and T.H. wrote the first draft of the paper and all authors (see list above) contributed to the subsequent drafts.

## Competing interests

The authors declare no competing interests.
