## [Peer Review File · Nature Communications]

Peer Review File - Reviewers' comments first round:

Reviewer #1 (Remarks to the Author):

General Comments:

The authors state they have used biomimicry to advance the restoration of salt marshes and seagrass meadows, by employing biodegradable mimics of below-and above-ground structures. To mention avionics, transportation and power generation as examples of the use of biomimicry in the context of the experiments described by the authors is far too great a stretch to even merit serious consideration. This technique uses more of the understanding of the habit of these foundational species and restoring that function to facilitate the restoration of the species- not to the benefit of another or to a [human] problem which is what biomimicry typically refers. And there is no mention of living shorelines which have been used to restore nature (using the authors take on biomimicry) until the Discussion (Lns 303-307) where it is mentioned as an area that would benefit- although many reef designs do exactly what the authors are attempting.

The notions that increased rhizome and shoot densities provided by artificial structures could enhance survival of seagrass and salt marsh species are what would be expected by experienced practitioners. Thus, there is nothing here that would not be predicted a priori.

One wonders if the authors really believe the results presented in this paper from a single temperate and tropical site for seagrasses and salt marshes are "...to pave the way for a new line of research studying how key, facilitative traits of organisms at point of establishment can mimicked to benefit ecological restoration for various environments". If so, it would be most useful to have been provided estimates of their costs of restoring unit areas of both seagrass meadows and salt marshes, for comparison with existing methods. That is, seagrass restoration is often very costly and infrequently successful. If the cost of the method proposed here was comparatively low, this could be an important selling point.

Given the sampling design issues (small replicate number and the confounding of latitude and seagrass and marsh species; we suggest asking a statistician for comment on the design)) and the non-surprising results obtained, a short report containing these results would be appropriate for a journal such as Restoration Ecology if the rhetoric on biomimicry were toned down. If information on costs were included, this could be a valuable full-length contribution to Restoration Ecology.

Specific Comments:

Ln 75- "Science and society HAS"

Ln 90 – remove "other" in front of services and replace with "the"

Figure 1 (d) – transplant is difficult to see. 1(e) – does the decomposing structure provide additional nutrients? It looks like there is a benthic bloom happening just within the area of where the mat was buried.

Lns 324-327 – It is well established that careful site selection for restoration activities is absolutely imperative for a successful outcome. Biomimicry would not be beneficial in the environments listed because the sites are not appropriate for restoration (disturbance is too intense, other stressors are still present).

Ln 325 "Physical disturbances are absent" -is this implying that there is no benefit to this technique unless there is some level of disturbance? Is this not benefiting restoration because you can use less material with the added benefit that it may expand the limit of physical disturbance the transplants can handle?

Specific comments (Results):

Ln 182 – control shoot counts for Bonaire seagrass is 0+0 but in the survival figure, there is ~20% survival? Not following how the shoot count would then be 0.

Lns 184 -185 –"numbers in the belowground structures were 53-times (0.9+1 shoots) and 2.6 times (2.6 + 0.8 shoots) times lower" - remove the second times at end of sentence.

Also these shoot counts are lower than what was initially transplanted (17.6 + 0.4 and 4.9+0.2), more so for the Netherlands than for Florida. It seems as though the belowground structures were not beneficial to the Netherlands cordgrass and if the experiment would have been extended, at either location, a more negative effect would have been seen.

Lns 96-197 – Remove irrespective of ecosystem type. There was no lateral expansion for the salt marsh controls because the transplant survival was 0%.

Lns 202-203 – Remove "Cordgrass expansion was zero in all controls" Again- no expansion

because the transplants did not survive.

Reviewer #2 (Remarks to the Author):

In this paper, the authors have conducted a distributed study using two ecosystem types (seagrass and salt marsh grass) and in both temperate and tropical locations to evaluate the utility of a potato-based mesh material matrix in restoration of these foundational species. They hypothesize that this material will help to stabilize sediment and reduce water flow, resulting in improved establishment and persistence of the target species. This is an ambitious and elegant coordinated study by talented researchers that very clearly shows the mesh aided in restoring the seagrasses *Zostera marina* and *Thalassia testudinum* through primarily sediment stabilization effects, and the cordgrasses *Spartina anglica* and *S. alterniflora*, primarily through aboveground flow reduction.

There are three main issues that I'd like to raise. First, the authors make a bit too much of biomimicry being new to restoration. Particularly in living shorelines type projects, biomimicry is central to restoration designs in many locations; e.g., creating substrata from shell or concrete that is full of shell material to simulate the structure and chemistry of natural oyster reefs, and for these restored reef structures to mimic natural oyster reefs in their ability to reduce wave energy. Econcrete and Reef Balls are two materials/ventures that attempt to simulate the physical structure of reefs, including hiding places for fish and invertebrates, built in tide pools, etc., and both are commonly used in restoration projects. 3-D printing of materials is also being used in many projects, such as in seawall panels that mimic the complexity of a rocky shore habitat or reef surface, thus increasing roughness that reduces wave energy and reflection and increases habitat value. I would recommend tempering the statements that humans have yet to leverage biomimicry to restore degraded ecosystems. There is some novelty in doing this for coastal plants, but even in these systems, mimicry is being used already; e.g., in Long Island Sound burlap planting disks through which eelgrass is inserted to stabilize sediment and plants as they establish, or buoy-deployed seeding, which takes flowering shoots and buoys them at a restoration site to simulate natural dispersal of rafting flowering shoots that will drop seed.

The second issue is that for the mesh technique to be up-scalable as the authors propose, we need to know something about the labor and cost of using the structures involved. There seemed to be a substantial amount of work to install the mesh structures, with rebar anchors, wooden poles and wiring. I also assume that a great deal of sediment disturbance is necessary to place the structures belowground. What spacing of the mesh structures would the authors propose and what would the costs be per acre? I wonder if this technique would be best used in small patches with wide spacing to establish nuclei of vegetation that can then facilitate further filling in of the plants.

Third, it would be nice to know what the flow rates are like in each of the experimental sites and how those compare to the range used in the flumes. The authors note "relatively exposed hydrodynamic conditions and mobile sediments" at their restoration sites but have not quantified this. It is surprising that the survivorship was so low for both cordgrass and seagrass in the controls, considering that the authors note these sites previously supported plants. Is the self-facilitation gained through the mesh structures needed only in places with high wave energy? More discussion and consideration of flow conditions that might warrant the mesh structures would help to put these results into context.

I really like this paper and support its publication in this journal if the authors can temper some statements and expand on others as described above.

[redacted]

Reviewer #3 (Remarks to the Author):

This paper examines the effects of erosion reduction techniques on salt marsh and seagrass restoration. The restoration approaches in this paper are good and the erosion problems they are trying to address are globally significant and growing. A more direct assessment of these erosion problems and a more nuanced approach to the novelty of the approaches would greatly improve the paper.

The most interesting results in this paper are not well developed and a bit buried beneath what seems like a lot of jargon on "trait-based biomimicry". The potentially new results in this paper are on the hydrodynamics of their erosion reduction measures and why above and below surface structures work better for restoring marshes and seagrasses respectively. I believe that these results from the field and flume should be a much greater focus of the work.

I have concerns that the whole focus on biomimicry mainly inflates the importance of the work and makes it appear as though the approaches are entirely novel; this has the effect of then ignoring a substantial literature on habitat restoration. Lines 244-259 seem particularly 'over the top' in the focus on biomimicry and the case for its novelty.

At a certain level, the whole, huge field of ecological restoration is already based on biomimicry. That is, the point of most ecological restoration is to mimic some of the biology/ecology that once existed in a place. The field of ecological restoration and the restoration literature is replete with examples of how natural, artificial and hybrid structures have been used to re-create physical conditions and/or ameliorate stresses such as erosion. The paper can argue with how well restoration ecologists have done and how much improvement is needed, but to largely ignore this past work by suggesting that the paper represents a new field/paradigm of biomimicry is not a helpful advancement.

More specifically, the paper does not acknowledge that the use of erosion control structures (including biodegradable structures) to enhance vegetation growth is widespread. Even construction contractors in my town use such practices (and these practices are also used in native plant restoration).

More specifically in the coastal and marine environment the use of "biological mimics" is widespread for example in oyster reef restoration with tens to hundreds of millions spent on oyster balls, castles, and blocks to enhance the bottom and improve oyster settlement and growth (even using techniques such as using oyster shell in the limestone to increase larval settlement as oysters 'smell' the bottom). Lines 296-310 acknowledge some of this work in wetlands and reefs but the connection (or lack thereof?) between this past work and the paper's 'first step' in biomimicry is not discussed.

The authors should step back a great deal from this focus on novel biomimicry and possibly discard this notion entirely. Then the paper can look more critically at the gaps that do exist. Erosion is a growing problem on coastlines in general and specifically for wetland restoration. There is good reason to acknowledge and to be critical of some of the approaches that are being used to reduce erosion in wetland restoration (e.g., across the work on 'living shorelines'). There is also a need to be critical of some of the wetland restoration that is failing because erosion is not addressed or acknowledged (though admittedly these 'failures' are not usually documented in the literature).

Reviewer #1 (Remarks to the Author):

General comments

1) *The authors state they have used biomimicry to advance the restoration of salt marshes and seagrass meadows, by employing biodegradable mimics of below-and above-ground structures. To mention avionics, transportation and power generation as examples of the use of biomimicry in the context of the experiments described by the authors is far too great a stretch to even merit serious consideration. This technique uses more of the understanding of the habit of these foundational species and restoring that function to facilitate the restoration of the species- not to the benefit of another or to a [human] problem which is what biomimicry typically refers.*

Reply: We agree that our statements regarding the novelty of biomimicry in restoration were overstated. Therefore, we removed this emphasis from paper and instead focused in more detail on the trait-based aspects as detailed in comment 5.

2) *And there is no mention of living shorelines which have been used to restore nature (using the authors take on biomimicry) until the Discussion (Lns 303-307) where it is mentioned as an area that would benefit- although many reef designs do exactly what the authors are attempting.*

Reply: We agree that we should have more explicitly contextualized our study within previous work. We therefore added references regarding earlier important work in relation to our study. Please refer to comment 4 for more details.

3) *The notions that increased rhizome and shoot densities provided by artificial structures could enhance survival of seagrass and salt marsh species are what would be expected by experienced practitioners. Thus, there is nothing here that would not be predicted a priori.*

Reply: Indeed, the facilitating effects of high stem densities and dense root mats on the established vegetation itself are well known. In fact, we also highlight this in the introduction as it forms an important basis for our work:

L142-146: Earlier observational and experimental work revealed that root mats of both seagrass and cordgrass are important for stabilizing sediment^{16,28,29,31,32}. Attenuation of hydrodynamic energy and resulting sediment accumulation by aboveground stems, on the other hand, is much stronger in patches of stiff salt marsh cordgrass stems compared to drag-avoiding, flexible seagrass shoots³³⁻³⁵.

- 16 Silliman, B. R. *et al.* Facilitation shifts paradigms and can amplify coastal restoration efforts. *Proceedings of the National Academy of Sciences* **112**, 14295-14300 (2015).
- 28 Maxwell, P. S. *et al.* The fundamental role of ecological feedback mechanisms for the adaptive management of seagrass ecosystems – a review. *Biological Reviews* **92**, 1521-1538 (2016).
- 29 Bouma, T. J. *et al.* Density-dependent linkage of scale-dependent feedbacks: a flume study on the intertidal macrophyte *Spartina anglica*. *Oikos* **118**, 260-268 (2009).
- 31 Christianen, M. J. A. *et al.* Low-canopy seagrass beds still provide important coastal protection services. *PLOS ONE* **8**, e62413 (2013).
- 32 Lo, V., Bouma, T., Van Belzen, J., Van Colen, C. & Airoldi, L. Interactive effects of vegetation and sediment properties on erosion of salt marshes in the Northern Adriatic Sea. *Marine environmental research* **131**, 32-42 (2017).
- 33 Bouma, T. J. *et al.* Organism traits determine the strength of scale-dependent bio-geomorphic feedbacks: A flume study on three intertidal plant species. *Geomorphology* **180-181**, 57-65 (2013).
- 34 Bouma, T. J. *et al.* Trade-offs related to ecosystem engineering: a case study on stiffness of emerging macrophytes. *Ecology* **86**, 2187-2199 (2005).

- 35 Peralta, G., Van Duren, L., Morris, E. & Bouma, T. Consequences of shoot density and stiffness for ecosystem engineering by benthic macrophytes in flow dominated areas: a hydrodynamic flume study. *Marine Ecology Progress Series* **368**, 103-115 (2008).

However, the inclusion of such positive interactions by altering restoration designs was only recently suggested by Silliman *et al.* (2015)¹⁶. Whereas earlier studies demonstrated that increased planting density can increase restoration success, the important novelty advanced by Silliman *et al.* (2015) was to not change density, but modify aggregation while keeping overall density unchanged. Consequently, restoration yields can increase by altering the design without increasing investments or the amount of donor material:

L104-112: A recent innovation in coastal restoration revealed that harnessing self-facilitation by altering restoration designs can increase restoration yields¹⁶. Whereas earlier work showed that increasing planting density can increase restoration success^{17,18}, Silliman *et al.* (2015) demonstrated that yields can be doubled simply by planting in clumps rather than applying commonly used plantation-style dispersed designs, while keeping overall density unchanged¹⁶. Although this simple clumping technique has the potential to fundamentally change coastal restoration^{12,19,20}, facilitation-harnessing approaches could become particularly effective if the organism traits generating self-facilitation can be mimicked and, thus, produced and distributed at large scales.

12. Gedan, K. & Silliman, B. *Patterns of salt marsh loss within coastal regions of North America: Presettlement to present.* (2009).
- 16 Silliman, B. R. *et al.* Facilitation shifts paradigms and can amplify coastal restoration efforts. *Proceedings of the National Academy of Sciences* **112**, 14295-14300 (2015).
- 17 van Katwijk, M. M. *et al.* Global analysis of seagrass restoration: the importance of large-scale planting. *Journal of Applied Ecology* **53**, 567-578 (2016).
- 18 Teas, H. J. Ecology and restoration of mangrove shorelines in Florida. *Environmental Conservation* **4**, 51-58 (1977).
- 19 Renzi, J. J., He, Q. & Silliman, B. R. Harnessing positive species interactions to enhance coastal wetland restoration. *Frontiers in Ecology and Evolution* **7**, doi:10.3389/fevo.2019.00131 (2019).
- 20 Shaver, E. C. & Silliman, B. R. Time to cash in on positive interactions for coral restoration. *PeerJ* **5**, e3499 (2017).

We build upon these findings to show that (1) the type of facilitation required is highly dependent on the traits of the species involved, and (2) mimicking these emergent traits rather than harnessing them by transplanting in clumped arrangements¹⁶ can minimize donor material requirements:

L112-114: Such innovation would eliminate the need for acquiring large numbers of transplants that may harm donor populations or require expensive nurseries.

L228-234: Our approach builds upon these efforts by experimentally demonstrating that tailor-made mimicry of species-specific key emergent traits – identified from past ecological studies – facilitates the establishment of different habitat-forming species. Specifically, our results highlight that by mimicking dense cordgrass patches that attenuate hydrodynamic energy^{35,46} or extensive seagrass root mats that improve sediment stability³¹, restoration success can be greatly enhanced and, in many cases, may turn failures into successes.

L253-257: Combined with the finding that belowground structures provide better sediment stabilization compared to aboveground treatments, these differences in stem traits explain the differential, ecosystem-specific results, highlighting the need to tailor emergent trait-based restoration approaches to specific habitat-forming species and environmental conditions.

L258-269: Recent experimental work from Dutch and US salt marshes demonstrates that harnessing beneficial species interactions through design can double restoration yields, because self-facilitation is instantaneously created by “clumping” transplants¹⁶. Although clumping into larger patches can enhance transplant survival, it diminishes the transplants’ potential to

expand laterally, because the relative edge length along which the vegetation can expand decreases isometrically with increasing patch size⁵⁰. Therefore, clumped configurations require more transplant units to achieve lateral outgrowth rates that sufficiently warrant recolonization. Here, we show that by deploying transplants inside establishment structures, our salt marsh transplant size was nine times smaller compared to the earlier applied “clumped” transplant design¹⁶, greatly reducing the need for donor material and avoiding potential damage to donor sites or demands on nurseries to cultivate transplants.

- 4) *One wonders if the authors really believe the results presented in this paper from a single temperate and tropical site for seagrasses and salt marshes are “..to pave the way for a new line of research studying how key, facilitative traits of organisms at point of establishment can mimicked to benefit ecological restoration for various environments”. If so, it would be most useful to have been provided estimates of their costs of restoring unit areas of both seagrass meadows and salt marshes, for comparison with existing methods. That is, seagrass restoration is often very costly and infrequently successful. If the cost of the method proposed here was comparatively low, this could be an important selling point.*

Reply: We agree that this statement was too bold, especially given the previous biomimicry focus. Therefore, we removed this focus from the manuscript, and instead emphasize more the trait-based restoration aspect of our study as its major novelty. Specifically, we show that tailor-made mimicry of previously identified species-specific emergent traits facilitates the establishment of different habitat-forming species, thereby minimizing costs and donor material requirements. Please see comment 5 for further details on this point.

Furthermore, we provide a more detailed explanation regarding the conditions under which we believe our approach is helpful (see comment 2, L281-292), and added cost estimates based on 4 scenarios (see comment 2, L275-281). We agree that these inclusions indeed greatly improved the manuscript.

Finally, we also down toned the sentence highlighted by the reviewer:

L323-326: Hence, we suggest that our trait-based approach may inspire a new research avenue investigating how mimicry of emergent traits by habitat-forming species may enhance establishment and restoration yields in harsh environments.

- 5) *Given the sampling design issues (small replicate number and the confounding of latitude and seagrass and marsh species; we suggest asking a statistician for comment on the design)) and the non-surprising results obtained, a short report containing these results would be appropriate for a journal such as Restoration Ecology if the rhetoric on biomimicry were toned down. If information on costs were included, this could be a valuable full-length contribution to Restoration Ecology.*

Reply: Indeed, as we investigated our restoration approach at both temperate and tropical sites to test its generality, seagrass and salt marsh plant species obviously also differ between sites. Although the different species share important traits (e.g. all plants form dense root mats; both seagrasses have flexible shoots; both salt marsh plants have stiff stems), we therefore analysed each site separately, thus yielding lower replicate numbers per ecosystem type (4 to 8 replicates per site). Despite this lower replication (which lowers statistical sensitivity), we nevertheless detect clear and significant outcomes, with great similarity between tropical and temperate sites due to the trait-specific (rather than species-specific) results.

To clarify our rationale in the manuscript text, we modified the methodological text dealing with the statistics according to comment 3, L414-416.

Clearly, we do not agree with the reviewer's advice regarding the target journal, given that we advance and experimentally test a new restoration approach across four ecosystems in two climate zones, with a potential applicability in many degraded ecosystems that depend on self-facilitation for persistence (e.g. coral and shellfish reefs, mangroves, rivers, peatlands, and (semi-)arid lands). We hope that reviewer 1 can share our opinion now that we have modified our conceptual focus and more precisely explained the rationale underlying our approach.

Specific Comments:

6) Ln 75- "*Science and society HAS*"

Reply: The line has been removed from the text.

7) Ln 90 – remove "*other*" in front of services and replace with "*the*"

Reply: We changed "*other*" with "*the*".

8) *Figure 1 (d) – transplant is difficult to see. 1(e) – does the decomposing structure provide additional nutrients? It looks like there is a benthic bloom happening just within the area of where the mat was buried.*

Reply: The transplant is now visible in Figure 1d.

The colour most likely emerged because the buried structures stabilized the sediment, thereby facilitating diatom growth (Barraquand *et al.* 2018).

Barraquand, F., Picoche, C., Maurer, D., Carassou, L., & Auby, I. (2018). Coastal phytoplankton community dynamics and coexistence driven by intragroup densitydependence, light and hydrodynamics. *Oikos*, 127(12), 1834-1852.

9) *Lns 324-327 – It is well established that careful site selection for restoration activities is absolutely imperative for a successful outcome. Biomimicry would not be beneficial in the environments listed because the sites are not appropriate for restoration (disturbance is too intense, other stressors are still present).*

Reply: We agree with the reviewer that site selection is key to success. When conditions are benign, or alternatively, the facilitation generated is insufficient to mitigate physical stressors, our approach is not applicable. In the first case, simple planting or seeding approaches may be sufficient and cheaper, while in the latter case engineering techniques (e.g. construction of dams) or a different site altogether may be more suitable approaches.

L289-292: By contrast, the approach is likely unsuitable for benign conditions, where seeding or dispersed transplant designs may prove to be more cost-efficient alternatives^{16,17,56} or when the environmental conditions are too harsh to be sufficiently mitigated by emergent traits of an established population.

10)Ln 325 "*Physical disturbances are absent*" -*is this implying that there is no benefit to this technique unless there is some level of disturbance? Is this not benefiting restoration because you can use less material with the added benefit that it may expand the limit of physical disturbance the transplants can handle?*

Reply: We believe that this approach is particularly useful in harsh environments where transplants benefit from facilitation. Here, facilitation broadens the organism's realized niche, thereby allowing the organism to thrive in conditions that without facilitation would not have been possible. In environments with low physical stress,

small transplants can survive without facilitation. In such conditions, our approach does not benefit survival and will merely increase restoration costs (material, labour, ..).

Please see comment 15 for textual changes.

Specific comments (Results):

11)Ln 182 – control shoot counts for Bonaire seagrass is 0+0 but in the survival figure, there is ~20% survival? Not following how the shoot count would then be 0.

Reply: The shoot number in Bonaire was not 0, but 0.25 shoots m⁻². We changed this in the manuscript.

12)Lns 184 -185 –“numbers in the belowground structures were 53-times (0.9+1 shoots) and 2.6 times (2.6 + 0.8 shoots) times lower” - remove the second times at end of sentence.

Reply: We removed the second “lower”.

13)Also these shoot counts are lower than what was initially transplanted (17.6 + 0.4 and 4.9+0.2), more so for the Netherlands than for Florida. It seems as though the belowground structures were not beneficial to the Netherlands cordgrass and if the experiment would have been extended, at either location, a more negative effect would have been seen.

Reply: We agree with the reviewer. In contrast to the aboveground structure in which the plants grow and expand, the belowground structure does not sufficiently facilitate cordgrass survival in the Netherlands and Florida to warrant long-term success. We now clarify this in the 2nd paragraph of the results:

L187-190: As these shoot numbers are below the initial count in the transplants (17.6 ± 0.4 and 4.9 ± 0.2 shoots/transplant in the Netherlands and Florida respectively), these results suggest that belowground structures do not sufficiently facilitate cordgrass to warrant long-term success.

14)Lns 96-197 – Remove irrespective of ecosystem type. There was no lateral expansion for the salt marsh controls because the transplant survival was 0%.

Reply: We removed “irrespective of ecosystem type”.

15)Lns 202-203 – Remove “Cordgrass expansion was zero in all controls” Again- no expansion because the transplants did not survive.

Reply: We added “because the transplants did not survive” to make this point clear.

L199-200: Cordgrass expansion was zero in all controls, because the transplants did not survive.

Reviewer #2 (Remarks to the Author):

16)In this paper, the authors have conducted a distributed study using two ecosystem types (seagrass and salt marsh grass) and in both temperate and tropical locations to evaluate the utility of a potato-based mesh material matrix in restoration of these foundational species. They hypothesize that this material will help to stabilize sediment and reduce water flow, resulting in improved establishment and persistence of the

target species. This is an ambitious and elegant coordinated study by talented researchers that very clearly shows the mesh aided in restoring the seagrasses *Zostera marina* and *Thalassia testudinum* through primarily sediment stabilization effects, and the cordgrasses *Spartina anglica* and *S. alterniflora*, primarily through aboveground flow reduction.

Reply: We were happy to read the referee's positive and helpful evaluation.

17) There are three main issues that I'd like to raise. First, the authors make a bit too much of biomimicry being new to restoration. Particularly in living shorelines type projects, biomimicry is central to restoration designs in many locations; e.g., creating substrata from shell or concrete that is full of shell material to simulate the structure and chemistry of natural oyster reefs, and for these restored reef structures to mimic natural oyster reefs in their ability to reduce wave energy. Econcrete and Reef Balls are two materials/ventures that attempt to simulate the physical structure of reefs, including hiding places for fish and invertebrates, built in tide pools, etc., and both are commonly used in restoration projects. 3-D printing of materials is also being used in many projects, such as in seawall panels that mimic the complexity of a rocky shore habitat or reef surface, thus increasing roughness that reduces wave energy and reflection and increases habitat value. I would recommend tempering the statements that humans have yet to leverage biomimicry to restore degraded ecosystems. There is some novelty in doing this for coastal plants, but even in these systems, mimicry is being used already; e.g., in Long Island Sound burlap planting disks through which eelgrass is inserted to stabilize sediment and plants as they establish, or buoy-deployed seeding, which takes flowering shoots and buoys them at a restoration site to simulate natural dispersal of rafting flowering shoots that will drop seed.

Reply: We agree that we overstated our biomimicry focus and we insufficiently acknowledged earlier work. Please see comments 4 and 5 for further details on how we dealt with these issues.

18) The second issue is that for the mesh technique to be up-scalable as the authors propose, we need to know something about the labor and cost of using the structures involved. There seemed to be a substantial amount of work to install the mesh structures, with rebar anchors, wooden poles and wiring. I also assume that a great deal of sediment disturbance is necessary to place the structures belowground. What spacing of the mesh structures would the authors propose and what would the costs be per acre? I wonder if this technique would be best used in small patches with wide spacing to establish nuclei of vegetation that can then facilitate further filling in of the plants.

Reply: We agree with the reviewers that inclusion of the cost of our approach has improved the manuscript. Please see comment 1 for details and Table S2 in the text for our cost summary.

Regarding the statement on the "substantial amount of work": In the Dutch salt marsh we, in retrospect, over-engineered our design (poles, rebars, and wiring to anchor the sheet down). At the other three sites we found that a simplified design using only rebar anchors and BESE sheets to construct the plots is sufficient, which of course substantially decreases time and costs.

Indeed, the amount of work involved for installing buried plots is clearly larger compared to constructing aboveground plots, but is nevertheless feasible. To highlight the differences in labour, we made a distinction between installation time and costs of seagrass (buried) and salt marsh (aboveground) plots in our cost calculation scenarios.

19) Third, it would be nice to know what the flow rates are like in each of the experimental sites and how those compare to the range used in the flumes. The authors note “relatively exposed hydrodynamic conditions and mobile sediments” at their restoration sites but have not quantified this. It is surprising that the survivorship was so low for both cordgrass and seagrass in the controls, considering that the authors note these sites previously supported plants. Is the self-facilitation gained through the mesh structures needed only in places with high wave energy? More discussion and consideration of flow conditions that might warrant the mesh structures would help to put these results into context.

Reply: We agree with the reviewer that a detailed description of hydrodynamic conditions would help to put our results into context, and enable comparisons to the flume results. Please see comment 2 for details on how we resolved this issue.

Additionally, the survivorship of cordgrass and seagrass transplants was probably low at these sites because the transplant size was too small to generate the level of self-facilitation required for survival. In the previous established vegetated state, vegetation patch sizes were likely much larger, generating self-facilitation (e.g. sediment stabilization, wave attenuation) required for individual plants to survive in these conditions. Natural establishment of the vegetation at such sites can only occur during a “Window of Opportunity”, a sufficiently long period of exceptionally calm conditions during which a recruit can establish. We have now included this explanation in the 3rd paragraph of the Introduction:

L126-135: However, to enable establishment under such conditions, a critical threshold for population size and/or density thus needs to be overcome²⁹. Under natural conditions, establishment may occur during a Window of Opportunity - a sufficiently long period of exceptionally calm conditions during which isolated individuals or small clones can settle and grow³⁰. However, such Windows are relatively rare and, as a consequence, natural re-establishment processes often take decades or longer. In such systems, restoration can act to accelerate this temporal delay by transplanting sufficiently large populations or clones¹⁶. However, transplantation at the required scale is often infeasible because of the resources and time required to harvest or cultivate, and then transplant sufficient material.

Here, we propose to address this limitation...

20) I really like this paper and support its publication in this journal if the authors can temper some statements and expand on others as described above.

Reply: Thank you very much for your constructive and thoughtful review.

Reviewer #3 (Remarks to the Author):

21) This paper examines the effects of erosion reduction techniques on salt marsh and seagrass restoration. The restoration approaches in this paper are good and the erosion problems they are trying to address are globally significant and growing. A more direct assessment of these erosion problems and a more nuanced approach to the novelty of the approaches would greatly improve the paper.

Reply: We are pleased to read this statement.

For our reply addressing erosion, please see comment 2.

22) The most interesting results in this paper are not well developed and a bit buried beneath what seems like a lot of jargon on “trait-based biomimicry”. The potentially new results in this paper are on the hydrodynamics of their erosion reduction measures and why above and below surface structures work better for restoring marshes and

seagrasses respectively. I believe that these results from the field and flume should be a much greater focus of the work.

Reply: We removed the biomimicry focus from the paper. Please see comment 5 for details.

We agree that a more detailed discussion on how our approach may aid in restoring degraded ecosystems along hydrodynamically exposed, and eroding shorelines benefits our manuscript. Therefore, we have revised the Discussion and updated the supplementary table regarding site conditions to include more details on hydrodynamic conditions. Please see comment 2 for more details.

23) I have concerns that the whole focus on biomimicry mainly inflates the importance of the work and makes it appear as though the approaches are entirely novel; this has the effect of then ignoring a substantial literature on habitat restoration. Lines 244-259 seem particularly 'over the top' in the focus on biomimicry and the case for its novelty. At a certain level, the whole, huge field of ecological restoration is already based on biomimicry. That is, the point of most ecological restoration is to mimic some of the biology/ecology that once existed in a place. The field of ecological restoration and the restoration literature is replete with examples of how natural, artificial and hybrid structures have been used to re-create physical conditions and/or ameliorate stresses such as erosion. The paper can argue with how well restoration ecologists have done and how much improvement is needed, but to largely ignore this past work by suggesting that the paper represents a new field/paradigm of biomimicry is not a helpful advancement.

Reply: As mentioned above, we agree regarding the prior focus on biomimicry and modified the manuscript accordingly.

We also agree that earlier work should be more thoroughly recognized and integrated. See comment 4 for details.

In addition, we provide information how we can optimize mimics and work towards better "restoration products":

L293-311: While our experimental results show that the establishment structures used here can enhance restoration success, and costs are such that upscaling is feasible, our mimicry of emergent traits is still relatively crude, highlighting a potential need for optimization. 3D-printing may, for example, prove a very useful tool to develop new biodegradable prototypes as it opens up virtually infinite design possibilities and allows for fine details at the micro-scale^{57,58}. To enable such optimization, identifying the bottlenecks that hamper establishment of the target species should be the first step^{19,53,59}. Next, it should be established whether the target species, or species that mutualistically interact with the target species⁶⁰, possesses emergent traits that mitigate these bottlenecks, after which the establishment structure's design can be improved to more accurately simulate these traits. In many cases, however, there may be multiple solutions to emulate a certain emergent trait, turning such a design optimization goal into a complex problem with many potential solutions, particularly when there are multiple traits to be considered. In engineering design, such a complex, 'wicked' problem is often approached using a morphological analysis that allows exploration of all possible solutions for the combinations of functions one aims to achieve⁶¹. For restoration, morphological analysis may help design structures that simultaneously ameliorate multiple emergent trait-mitigated bottlenecks, such as wave attenuation combined with sediment stabilization by coastal vegetation, or provisioning of attachment substrate combined with predation shelter by oysters and mussels^{28,62-67}.

24) More specifically, the paper does not acknowledge that the use of erosion control structures (including biodegradable structures) to enhance vegetation growth is widespread. Even construction contractors in my town use such practices (and these practices are also used in native plant restoration).

Reply: We agree and now include previous work to solve this issue (see above).

25) *More specifically in the coastal and marine environment the use of “biological mimics” is widespread for example in oyster reef restoration with tens to hundreds of millions spent on oyster balls, castles, and blocks to enhance the bottom and improve oyster settlement and growth (even using techniques such as using oyster shell in the limestone to increase larval settlement as oysters ‘smell’ the bottom). Lines 296-310 acknowledge some of this work in wetlands and reefs but the connection (or lack thereof?) between this past work and the paper’s first step’ in biomimicry is not discussed.*

Reply: We agree and now include previous work on this point (see above).

26) *The authors should step back a great deal from this focus on novel biomimicry and possibly discard this notion entirely. Then the paper can look more critically at the gaps that do exist. Erosion is a growing problem on coastlines in general and specifically for wetland restoration. There is good reason to acknowledge and to be critical of some of the approaches that are being used to reduce erosion in wetland restoration (e.g., across the work on ‘living shorelines’). There is also a need to be critical of some of the wetland restoration that is failing because erosion is not addressed or acknowledged (though admittedly these ‘failures’ are not usually documented in the literature).*

Reply: We removed biomimicry from the paper entirely. Please see comment 5 for details.

For our reply addressing erosion, please see comment 2.

Reviewers' comments second round:

Reviewer #1 (Remarks to the Author):

The authors have made substantial changes to the overall benefit of the manuscript. The removal of the biomimicry narrative and replacement with emergent traits strengthens the overall manuscript. The concept of using biodegradable material to replace a facilitative trait and reduce the size of restoration plugs/transplants is one that furthers the restoration practice and is exciting to see being done.

The suggestion to publish in a journal such as Restoration Ecology was mentioned as a way to get this novel improvement in restoration design in the front of managers and practitioners as quickly as possible. Not to say that Nature Communications is not an outstanding journal, but not one a practitioner/ manager would necessarily look to first when researching new techniques that improve restoration success.

Other than a few edits for clarity (see below) we recommend accepting the manuscript.

Edits: Terminology that requires correcting is the repeated reference to the Florida salt marsh site as tropical. Clearly, this site between 29 and 30 degrees N is outside the tropics. Moreover, salt marshes are replaced in the tropics by mangroves so it is hard to understand why the authors would use the tropical designation for a warm temperate salt marsh. This means that any reference to the Florida salt marsh site as tropical must be changed.

Ken Heck and Dottie Byron

Reviewer #2 (Remarks to the Author):

This is an interesting and definitely publishable study, with an ambitious replication of a restoration tool (used above and belowground to reduce flow or stabilize sediments, respectively) across several locations and species of cordgrass and seagrass. With the exception of a few specific comments I'll detail below, the data presentation was well-done, the manuscript clearly written, and the nuances of the results interesting. The study shows methods of sediment stabilization may be most important to seagrasses (especially *Z. marina* in Sweden, less compelling for *T. testudinum* in Bonaire; see below) and those that reduce flow may be more important to cordgrasses (more compelling in the Netherlands than Florida; see below), in both cases at sites where recruitment with more standard planting techniques had very poor success. These are useful results and suggest that the focus for emergent versus submerged plants should be different when attempting to jumpstart restoration. I do find it to be a bit of stretch to say that mimicry of the traits of habitat engineering by foundational species is a major advancement to the field of ecological restoration, as restoration practitioners have long employed mimicry on the ground. I wonder if the authors are only aware of published literature, as admittedly, much of the restoration work and outcomes is not published or is in the grey literature; that does not mean that practitioners have not been employing such techniques (e.g., using recycled oyster shell to stimulate oyster settlement as in natural reefs, building perches to attract birds that will disperse seeds before there is tall vegetation, etc.). Thus, I find this to be a solid study but I don't think it is as novel as the authors claim.

A few more detailed comments:

I was confused by the stats description in the methods that does not appear to match the chi² results in the figure captions for Figs 2-4. A df of 2 seems to refer to the three levels of one treatment and no blocking, even though a blocked design is described in the methods. The paper could benefit from a stats table in the supplemental material (unless I just missed it somehow but I tried to go through everything) with which to ascertain how the results in the figure captions come from the stats that were described in the methods.

I appreciate the cost analysis but do wonder about burial of a textile on a large scale for seagrass restoration sites as it would lead to siltation in any but the most sandy sites, triggering additional permitting in many places. Further, .7 hours per buried plot is a long time if that is referring to a m2 plot, considering this work is dependent on wading/snorkeling on only the lowest tides (if these are subtidal plots as indicated) or doing all the work on SCUBA. The length of time required to accomplish the technique (in weeks or months) would be helpful to know if practitioners were to only be able to use the lowest tides during daylight or the extra cost per hour of SCUBA should be included.

Although the buried mats had significant effects on shoot numbers at both seagrass sites, the number of shoots only approximately doubled (from the initial 8 to about 16) in Bonaire after nearly two years. The shoot number may be significantly different from that of the no-mat controls, but does not seem like strong justification for the expense and time of the buried mats. The Swedish results for *Z. marina* are more compelling.

For both seagrasses and cordgrasses, change from initial would be better to show than simple counts of shoots; the number of cordgrass shoots in the Netherlands was fewer than initial at 3 of seven plots after two growing seasons which would be more apparent if change was used. This would also be much better for the Florida cordgrass which had approximately the same number as initial after 1 year with the aboveground mat and was lower than initial with belowground mats. Yes, this is greater than for no mats, but is that enough to justify the method?

A minor point but northern Florida would be a subtropical location, not tropical, right?

And I thought *Spartina anglica* was an invasive hybrid in Europe, but perhaps it is still a restoration target since it is naturalized at this point?

Reviewer #3 (Remarks to the Author):

This paper examines the effects of erosion reduction techniques on salt marsh and seagrass restoration. The restoration approaches in this paper are good and the erosion problems they are trying to address are globally significant and growing. The revision has addressed many of the difficult conceptual issues from the last version and does a better job focusing on the experiments and their results.

The abstract however still needs improvement to focus more specifically on the results and their tangible implications. More than half of the abstract is devoted to general theorizing.

I think the work is interesting and a useful contribution to restoration practice, but I do not find the results nearly as novel as the paper purports. This contribution does not represent "a new restoration concept" (line 135), which is the critical line of argument in the last paragraph of the introduction. The field of ecological restoration has many examples of how natural, artificial and hybrid structures have been used to re-create physical conditions and/or ameliorate stresses such as erosion.

In the last paragraph of the discussion (line 312-326) the paper does a better job of acknowledging the breadth of this prior restoration work. Nonetheless it appears to do so in a way that suggests the prior work somehow missed the point that controlling waterflow and erosion (and with semi-natural structures) can be important. I do not think this is case. The importance of using structures that help in early establishment by creating stable environments is well known while of course noting that these practices still can be improved.

I think this paper should address how this work relates to some of the more intensive approaches that are being used to reduce erosion in wetland restoration (e.g., on 'living shorelines'). In many coastal environments we may be past the point where restoration ecologists can use semi-natural structures such as mats to reduce erosion. Indeed in a growing number of contexts much more

aggressive approaches are necessary and being taken to reduce wave impacts and erosion using barriers (from semi-natural to entirely artificial). It is hard to address the topic of wetland restoration and erosion without addressing these more intensive examples which likely now number a hundred or more.

The paper now includes a table and section on costs and cost effectiveness. This is a promising start but it does not yet come together in a coherent way, which is probably why these results are identified in the Discussion. This case for cost effectiveness should be either solidified and put in the Methods/Results or dropped. The paper sets up an argument that (coastal) restoration is expensive and there is a strong need for more cost-effective approaches. This is true and well documented by others with direct data on costs for different approaches including different (semi) natural and artificial alternatives. Table S2 posits costs for the different approaches in this paper but in an admittedly "arbitrary" way (line S42). The paper could be better served by using the actual costs and rates of success from the experiments and extrapolating from them. Other recent papers do a better job of describing and reviewing actual costs of restoration.

These current 'cost effectiveness' values appear to be central to the main conclusion of the paper as indicated by the last sentence of the abstract: "Mimicking key emergent traits may allow upscaling of restoration in many ecosystems that depend on self-facilitation for persistence, by constraining biological material requirements and implementation costs."

This statement is not sufficiently supported in the manuscript in terms of upscaling or cost effectiveness.

Response letter to NCOMMS-19-7936145A

Dear editor and reviewers,

We wish to thank the reviewers once more for their helpful comments on our manuscript.

Following the suggestions of reviewers 2 and 3, we strengthened our cost-effectiveness analysis by providing a detailed description in the Methods and integrating the outcomes into the Results. Furthermore, we removed any statements that overemphasized the novelty of our work.

In addition to these most important changes, we addressed all other concerns and provide a detailed, point-by-point response on how we dealt with these issues below.

We hope that we addressed all comments and suggestions to your satisfaction.

Sincerely,

Tjisse van der Heide, on behalf of all authors

Reviewer #1 (Remarks to the Author):

1) The authors have made substantial changes to the overall benefit of the manuscript. The removal of the biomimicry narrative and replacement with emergent traits strengthens the overall manuscript. The concept of using biodegradable material to replace a facilitative trait and reduce the size of restoration plugs/transplants is one that furthers the restoration practice and is exciting to see being done.

The suggestion to publish in a journal such as Restoration Ecology was mentioned as a way to get this novel improvement in restoration design in the front of managers and practitioners as quickly as possible. Not to say that Nature Communications is not an outstanding journal, but not one a practitioner/ manager would necessarily look to first when researching new techniques that improve restoration success.

Other than a few edits for clarity (see below) we recommend accepting the manuscript.

Reply: We are happy to read the reviewers positive recommendation and helpful edits.

2) Edits: Terminology that requires correcting is the repeated reference to the Florida salt marsh site as tropical. Clearly, this site between 29 and 30 degrees N is outside the tropics. Moreover, salt marshes are replaced in the tropics by mangroves so it is hard to understand why the authors would use the tropical designation for a warm temperate salt marsh. This means that any reference to the Florida salt marsh site as tropical must be changed.

Reply: We agree that the saltmarsh in Florida should be described as subtropical. We changed the text, tables and figures accordingly.

Ken Heck and Dottie Byron

Reviewer #2 (Remarks to the Author):

3) This is an interesting and definitely publishable study, with an ambitious replication of a restoration tool (used above and belowground to reduce flow or stabilize sediments, respectively) across several locations and species of cordgrass and seagrass. With the exception of a few specific comments I'll detail below, the data presentation was well-done, the manuscript clearly written, and the nuances of the results interesting.

*The study shows methods of sediment stabilization may be most important to seagrasses (especially *Z. marina* in Sweden, less compelling for *T. testudinum* in Bonaire; see below) and those that reduce flow may be more important to cordgrasses (more compelling in the Netherlands than Florida; see below), in both cases at sites where recruitment with more standard planting techniques had very poor success.*

These are useful results and suggest that the focus for emergent versus submerged plants should be different when attempting to jumpstart restoration. I do find it to be a bit of stretch to say that mimicry of the traits of habitat engineering by foundational species is a major advancement to the field of ecological restoration, as restoration practitioners have long employed mimicry on the ground.

I wonder if the authors are only aware of published literature, as admittedly, much of the restoration work and outcomes is not published or is in the grey literature; that does not mean that practitioners have not been employing such techniques (e.g., using recycled oyster shell to stimulate oyster settlement as in natural reefs, building perches to attract birds that will disperse seeds before there is tall vegetation, etc.). Thus, I find this to be a solid study but I don't think it is as novel as the authors claim.

Reply: We thank the reviewer for his/her positive remarks. We are indeed aware that much of the restoration work and outcomes are not published in scientific journals, but can be found in grey literature or are shared more informally among restoration practitioners. In addition, the reviewer is of course correct that other restoration approaches in effect mimic certain traits of habitat-forming species to help support

restoration outcomes. However, we are not aware of any examples that have articulated and then tested this across different ecosystems types and geographic locations as a general framework for restoration that focuses on the “mimicry of emergent traits”. Thus, we feel the novelty of our work relative to prior restoration studies lies in both our contribution to communicating this general framework and examining its applicability in different contexts. We do agree that we unnecessarily overemphasized the novelty aspect of our approach in the Introduction and Discussion, and have therefore toned this down accordingly (see crossed-out text below):

Introduction

L135-137: Here, we propose to address this limitation and investigate a ~~new~~ restoration concept, inspired by recent advancements in transplant designs¹⁶ and based on engineering, in which we mimic key emergent traits that generate self-facilitation.

L137-140: We developed ~~novel~~ biodegradable establishment structures with the aim to enhance the survival and growth of small salt marsh grass and seagrass transplants (**Figure 1, S1**), thereby minimizing costs and the need for often-limited donor material.

Discussion

L338-341: Hence, we suggest that our trait-based approach may inspire a ~~new~~ follow-up research ~~avenue~~ investigating how mimicry of emergent traits by habitat-forming species may enhance establishment and restoration yields in harsh environments.

Please also note that in the previous revision, we already added a suite of literature describing related restoration techniques used by practitioners:

- 36 Wolters, M., Bakker, J. P., Bertness, M. D., Jeffries, R. L. & Möller, I. Saltmarsh erosion and restoration in south-east England: squeezing the evidence requires realignment. *Journal of Applied Ecology* **42**, 844-851 (2005).
- 37 Currin, C. A., Chappell, W. S. & Deaton, A. Developing alternative shoreline armoring strategies: the living shoreline approach in North Carolina. (2010).
- 38 Herbert, D. *et al.* Mitigating erosional effects induced by boat wakes with living shorelines. *Sustainability* **10**, 436 (2018).
- 39 Meyer, D. L., Townsend, E. C. & Thayer, G. W. Stabilization and erosion control value of oyster cultch for intertidal marsh. *Restoration Ecology* **5**, 93-99 (1997).
- 40 Baine, M. Artificial reefs: a review of their design, application, management and performance. *Ocean & Coastal Management* **44**, 241-259 (2001).
- 41 Graham, P. M., Palmer, T. A. & Beseres Pollack, J. Oyster reef restoration: substrate suitability may depend on specific restoration goals. *Restoration Ecology* **25**, 459-470 (2017).
- 42 Bersosa Hernández, A. *et al.* Restoring the eastern oyster: how much progress has been made in 53 years? *Frontiers in Ecology and the Environment* **16**, 463-471 (2018).
- 43 Spieler, R. E., Gilliam, D. S. & Sherman, R. L. Artificial substrate and coral reef restoration: what do we need to know to know what we need. *Bulletin of Marine Science* **69**, 1013-1030 (2001).

- 44 Morgan, R. P. & Rickson, R. J. *Slope stabilization and erosion control: a bioengineering approach*. (Taylor & Francis, 2003).
- 45 Suykerbuyk, W. *et al.* Unpredictability in seagrass restoration: analysing the role of positive feedback and environmental stress on *Zostera noltii* transplants. *Journal of Applied Ecology* 53, 774–784 (2016).

A few more detailed comments:

4) I was confused by the stats description in the methods that does not appear to match the chi2 results in the figure captions for Figs 2-4. A df of 2 seems to refer to the three levels of one treatment and no blocking, even though a blocked design is described in the methods. The paper could benefit from a stats table in the supplemental material (unless I just missed it somehow but I tried to go through everything) with which to ascertain how the results in the figure captions come from the stats that were described in the methods.

Reply: We agree with the reviewer that a statistics table is helpful, which we therefore added to the supplementary information. Indeed, a model with a normal distribution can be adjusted for random effects with e.g. Satterthwaite or Kenward-Roger corrections of *df*. While this was possible for sediment movement data, survival (binary) and shoot number (counts) instead followed binomial and Poisson distributions, respectively. Here, we used Wald Chi-square tests to assess treatment effects. Maximum lateral expansion did not match any distribution (even after transformation), and was therefore tested non-parametrically.

Table S3. Summary of all statistical results. Numbers indicate the statistical test used: ⁽¹⁾ = GLM with binomial distribution, ⁽²⁾ = GLMM with Poisson distribution and block effect, ⁽³⁾ = Kruskal-Wallis, ⁽⁴⁾ = t-test with unequal variances, and ⁽⁵⁾ = LMM with block effect.

Treatment	Variable	df	Chi- ⁽²⁾ , t- ⁽⁴⁾ or F-value ⁽⁵⁾	p -value
The Netherlands, temperate, Spartina anglica				
Aboveground	Survival ¹	2	22.17	<0.001
Belowground				
Control				
The Netherlands, temperate, Spartina anglica				
Aboveground	Shoot number ²	2	684.56	<0.001
Belowground				
Control				
The Netherlands, temperate, Spartina anglica				
Aboveground	Maximum lateral expansion ³	2	7.69	<0.05
Belowground				
Control				
Florida, USA, subtropical, Spartina alterniflora				
Aboveground	Survival ¹	2	15.28	<0.001
Belowground				
Control				

Florida USA, subtropical, Spartina alterniflora				
Aboveground	Shoot number ²	2	48.03	<0.001
Belowground				
Control				
Florida USA, subtropical, Spartina alterniflora				
Aboveground	Maximum lateral expansion ³	2	10.57	<0.01
Belowground				
Control				
Sweden, temperate, Zostera marina				
Aboveground	Survival ¹	2	6.28	<0.05
Belowground				
Control				
Sweden, temperate, Zostera marina				
Aboveground	Shoot number ²	2	33.3	<0.001
Belowground				
Control				
Sweden, temperate, Zostera marina				
Aboveground	Maximum lateral expansion ³	2	6.59	<0.05
Belowground				
Control				
Bonaire, tropical, Thalassia testudinum				
Aboveground	Survival ¹	2	6.28	<0.05
Belowground				
Control				
Bonaire, tropical, Thalassia testudinum				
Aboveground	Shoot number ²	2	28	<0.001
Belowground				
Control				
Bonaire, tropical, Thalassia testudinum				
Aboveground	Maximum lateral expansion ³	2	8.64	<0.05
Belowground				
Control				
Wave flume experiment with cordgrass mimics				
Aboveground	Shoot movement ⁴	1	-3.8758	<0.01
Control				
Seagrass sites (sediment movement)				
Structure	Sediment movement ⁵	2.1	22.392	<0.001
Location		1.6	24.015	<0.01

5) I appreciate the cost analysis but do wonder about burial of a textile on a large scale for seagrass restoration sites as it would lead to siltation in any but the most sandy sites, triggering additional permitting in many places. Further, .7 hours per buried plot is a long time if that is referring to a m2 plot, considering this work is dependent on wading/snorkeling on only the lowest tides (if these are subtidal plots as indicated) or doing all the work on SCUBA. The length of time required to accomplish the technique (in weeks or months) would be helpful to know if practitioners were to only be able to use the lowest tides during daylight or the extra cost per hour of SCUBA should be included.

Reply: We agree that we should have specified scuba costs in more detail in our estimates. Scuba equipment is required for installing structures at water depths below 0.5 m, which of course takes more preparations. However, it is overall not necessarily more time consuming, as the work is independent of the tidal cycle. Regardless, the burial of establishment structures is indeed the most tedious part of the work (30 minutes per m²). To prevent an underestimation of the costs, we made a rather conservative estimate based upon our own experiences. This estimate includes both construction costs (including scuba) and organisation/project planning time. For clarity, we now split the construction costs and organisation time. The updated calculations show that the use of scuba and boat fees increases the costs by 20% compared to intertidal work (see Table S2).

6) *Although the buried mats had significant effects on shoot numbers at both seagrass sites, the number of shoots only approximately doubled (from the initial 8 to about 16) in Bonaire after nearly two years. The shoot number may be significantly different from that of the no-mat controls, but does not seem like strong justification for the expense and time of the buried mats. The Swedish results for Z. marina are more compelling.*

Reply: Indeed, shoot numbers at both seagrass sites differ significantly between buried mat and control treatment, with the effect in absolute shoot numbers being the greatest in Sweden. However, we do not agree that the results from Sweden can therefore simply be interpreted as “more compelling” because, apart from the sites, also the species growing at these sites differ. Specifically, we used *Zostera marina* in Sweden, which is a much faster growing species than *Thalassia testudinum*, the species used in Bonaire. To clarify this difference, we now contrast the growth rates of both species in the Results and Discussion:

Results

L179-181: Seagrass shoot numbers were highest in belowground structures with 30.1 ± 5 shoots for *Z. marina* in Sweden and 15.5 ± 2 shoots for the slower-growing climax species *T. testudinum* in Bonaire.

Discussion

L264-266: In addition, the increase in shoot number differed considerably depending on whether a faster (e.g. *Z. marina*) or slower growing (e.g. *T. testudinum*) species was introduced.

L294-297: Our results highlight that under harsh conditions where self-facilitation is important, mimicry of self-facilitating, emergent traits can increase both restoration success, and cost-effectiveness, particularly when using fast-growing species and accepting a long restoration period (Table S2).

7) *For both seagrasses and cordgrasses, change from initial would be better to show than simple counts of shoots; the number of cordgrass shoots in the Netherlands was fewer than initial at 3 of seven plots after two growing seasons which would be more apparent if change was used. This would also be much better for the Florida cordgrass which had approximately the same number as initial after 1 year with the aboveground mat and was lower than initial with belowground mats. Yes, this is greater than for no mats, but is that enough to justify the method?*

Reply: We agree that comparing against initial shoot numbers, for instance by calculating absolute or relative changes, would show changes since the onset more clearly. However, such a representation would instead obscure the shoot numbers present on the plots at the end. Indeed, shoot numbers in the Netherlands were higher compared to Florida, but the Dutch transplants also grew for two growing seasons instead of one. In addition, transplants typically lose shoots initially due to transplantation stress. This was a relatively important factor at the Florida site, which is particularly stressful due to frequent disturbances from boat-generated waves (Herbert et al., 2018, Silliman et al. 2019). Nevertheless, despite shoot losses in the original plug, most transplants survived the winter (~75% in above- and belowground treatments; Fig. 2), recovered and grew. This is most clearly illustrated by the lateral expansion, which was in fact more rapid in Florida than in the Netherlands. Hence, we overall feel it is more accurate to depict the actual observed shoot numbers, particularly when assessing them in concert with transplant survival and lateral expansion, as we do here.

8) *A minor point but northern Florida would be a subtropical location, not tropical, right?*

Reply: We agree with the reviewer. As described in our response to reviewer 1 (see comment 2), we have made changes accordingly.

9) *And I thought *Spartina anglica* was an invasive hybrid in Europe, but perhaps it is still a restoration target since it is naturalized at this point?*

Reply: *Spartina anglica* is a cordgrass species that originated in England in about 1870, and is now considered a nonnative species endemic to Britain and the European

mainland. It was actively introduced in salt marshes throughout Europe in 1926 (the Netherlands), 1927 (Germany) and 1931 (Denmark). Thus, although this species is considered undesirable in some cases, it has also become an important marsh building species at many sites and is therefore often used in salt marsh restoration projects (e.g. Bakker *et al.*, 2002, Cao *et al.*, 2018).

Cao, H.; Zhu, Z.; Balke, T; Zhang, L.; Bouma, T.J.(2018). Effects of sediment disturbance regimes on *Spartina* seedling establishment: Implications for salt marsh creation and restoration. *Limnol. Oceanogr.* 63(2): 647-659.

Bakker, J. P., Esselink, P., Dijkema, K. S., Van Duin, W. E., & De Jong, D. J. (2002). Restoration of salt marshes in the Netherlands. *Hydrobiologia*, 478(1-3), 29-51.

Reviewer #3 (Remarks to the Author):

10) This paper examines the effects of erosion reduction techniques on salt marsh and seagrass restoration. The restoration approaches in this paper are good and the erosion problems they are trying to address are globally significant and growing. The revision has addressed many of the difficult conceptual issues from the last version and does a better job focusing on the experiments and their results.

Reply: We thank the reviewer for his/her positive comments.

11) The abstract however still needs improvement to focus more specifically on the results and their tangible implications. More than half of the abstract is devoted to general theorizing.

Reply: The current abstract is just below the journals word limit (148/150). In its current form, we first describe the background and theory (both 1 sentence), followed by our approach (1 sentence), results (2 sentences) and conclusions (1 sentence). We would prefer to leave the abstract unaltered as we feel it is rather well balanced. However, we are of course willing to make changes if the editor requests this.

12) I think the work is interesting and a useful contribution to restoration practice, but I do not find the results nearly as novel as the paper purports. This contribution does not represent "a new restoration concept" (line 135), which is the critical line of argument in the last paragraph of the introduction. The field of ecological restoration has many

examples of how natural, artificial and hybrid structures have been used to re-create physical conditions and/or ameliorate stresses such as erosion.

Reply: We have toned down our emphasis on novelty in the Introduction and Discussion. Please see comment 3 for details on how we resolved this issue.

13) In the last paragraph of the discussion (line 312-326) the paper does a better job of acknowledging the breadth of this prior restoration work. Nonetheless, it appears to do so in a way that suggests the prior work somehow missed the point that controlling waterflow and erosion (and with semi-natural structures) can be important. I do not think this is case. The importance of using structures that help in early establishment by creating stable environments is well known, while of course noting that these practices still can be improved.

Reply: We of course agree that previous restoration work took erosion and hydrodynamic conditions into consideration, and also explicitly acknowledge this in the first paragraph of the Discussion since the previous version:

L235-239: At present, erosion is an increasing problem along coastlines in general, and at degraded sites that require restoration in particular³⁶. To combat this pervasive challenge, hard structures from shells or concrete are often applied to provide stable substrates necessary to stimulate reef formation³⁷⁻⁴³, while sediment stabilisation measures have been used to support vegetation establishment^{44,45}.

However, to further emphasize this, we now cite two more papers in the last paragraph of the Discussion:

L322-326: For restoration, morphological analysis may help design structures that simultaneously ameliorate multiple emergent trait-mitigated bottlenecks, such as wave attenuation combined with sediment stabilization by coastal vegetation^{38,62}, or provisioning of attachment substrate combined with predation shelter by oysters and mussels^{28,38,63-68}.

³⁸Herbert, D. *et al.* Mitigating erosional effects induced by boat wakes with living shorelines. *Sustainability* **10**, 436 (2018).

⁶²Narayan, S. *et al.* The Effectiveness, Costs and Coastal Protection Benefits of Natural and Nature-Based Defences. *PLOS ONE* **11**, e0154735, doi:10.1371/journal.pone.0154735 (2016).

14) *I think this paper should address how this work relates to some of the more intensive approaches that are being used to reduce erosion in wetland restoration (e.g., on 'living shorelines'). In many coastal environments we may be past the point where restoration ecologists can use semi-natural structures such as mats to reduce erosion. Indeed, in a growing number of contexts much more aggressive approaches are necessary and being taken to reduce wave impacts and erosion using barriers (from semi-natural to entirely artificial). It is hard to address the topic of wetland restoration and erosion without addressing these more intensive examples, which likely now number a hundred or more.*

Reply: We agree that erosion in coastal restoration is a severe problem and, depending on the exposure level, can possibly only be ameliorated by hard defences. Rather than relying on hard structures for an indefinite period of time, our work advances an approach in which we temporarily mimic mature vegetation stands, thereby allowing small transplants to grow and establish. Once established, the restored system should function naturally, rendering the mimics obsolete. Hard defences may indeed become a solution in cases where conditions are too harsh to be sufficiently mitigated by emergent traits of an established population, and restoration would nevertheless be desired. Although we feel that an in-depth discussion on hard structures is beyond the scope of our paper, we now do address this option in the Discussion:

L301-307: This illustrates that trait-based mimicry design may be particularly helpful in harsh conditions where restoration is inherently failure-prone and expensive. By contrast, the approach is likely unsuitable for benign conditions, where seeding or dispersed transplant designs may prove to be more cost-efficient alternatives^{16,17,56} or when the environmental conditions are too harsh to be sufficiently mitigated by emergent traits of an established population. In the latter case, only permanent protection measures, such as hard defense structures, would provide a long-term feasible option to allow vegetation development.

15) *The paper now includes a table and section on costs and cost effectiveness. This is a promising start but it does not yet come together in a coherent way, which is probably why these results are identified in the Discussion. This case for cost effectiveness should be either solidified and put in the Methods/Results or dropped. The paper sets up an argument that (coastal) restoration is expensive and there is a strong need for more cost-effective approaches. This is true and well documented by others with direct data on costs for different approaches including different (semi)natural and artificial alternatives. Table S2 posits costs for the different approaches in this paper but in an*

admittedly “arbitrary” way (line S42). The paper could be better served by using the actual costs and rates of success from the experiments and extrapolating from them. Other recent papers do a better job of describing and reviewing actual costs of restoration.

Reply: Our main aim was to advance and test trait-based mimicry as a general restoration concept. Clearly, the biodegradable establishment structures used here are not the only feasible technique to fit this framework. Restoration costs based on trait-based mimicry will therefore vary depending on the target system and the specifics of the method used. Hence, the cost feasibility calculations from this study merely serve as an example to illustrate general applicability. With regard to costs, we did indeed use the data from our own experiments and extrapolated these in our calculations. For the expansion rates, we used a combination of our own data combined with those from other studies as we feel that the inclusion of extra data yields a more realistic estimate.

We agree with the reviewer that the above considerations were not yet explained clearly enough in the previous version. Therefore, as suggested by the reviewer, we now include a detailed description of our calculations in the Methods, and describe the outcomes in the Results section:

Methods

Cost feasibility analysis

L445-461: To illustrate the potential ‘real-world’ applicability of trait-based mimicry, we calculated construction costs for a number of scenarios in which we upscale our specific technique as an example. Specifically, we considered the following four scenarios for both seagrass and salt marshes: 1) short recovery time, fast plant growth, 2) long recovery time, fast plant growth, 3) short recovery time, slow plant growth, and 4) long recovery time, fast plant growth. We chose these specific scenarios because they reflect the trade-off between construction costs, species selection, and restoration time that restoration practitioners may face when applying this method. Based on actual restoration projects¹⁵, we chose two restoration periods in which complete recovery should be accomplished; i.e. 5 (short) vs. 10 (long) years to establish a continuous vegetation stand. In addition, we selected two contrasting lateral extension rates of transplants (i.e., fast vs. slow growth) to illustrate the effect of species selection on the costs. Construction costs are extrapolated from actual costs in our experiments. Lateral extension rates are based on data from this work, combined with additional data from literature^{16,78-80} (Figure 4, Table S2). In each scenario, the 1-m² establishment structures were assumed to be spread out evenly across space. Their required initial

cover (% of a hectare) depends on the selected restoration period and expansion rate of plant species.

Results

L218-227: To illustrate the potential scalability of trait-based mimicry as a general approach, we calculated construction costs for four scenarios per ecosystem in which we upscale our specific technique as an example. The costs to restore vegetated coastal ecosystems ranges from 5,000 to 280,000 US\$/ha (Table S2), depending on the plant expansion rate and the restoration period (5 or 10 years). For instance, using fast growing species and a long restoration period results in lowest costs with 6,250 and 5,000 US\$/ha for salt marsh and seagrass systems, respectively (Table S2). Costs increase 4-times to 25,000 and 20,000 US\$/ha when shortening the restoration period to 5 years. Selecting slow growing species and using a short restoration period, results in the highest costs of 100,000 and 280,000 US\$/ha for salt marsh and seagrass systems, respectively (Table S2).

16) These current 'cost effectiveness' values appear to be central to the main conclusion of the paper as indicated by the last sentence of the abstract: "Mimicking key emergent traits may allow upscaling of restoration in many ecosystems that depend on self-facilitation for persistence, by constraining biological material requirements and implementation costs." This statement is not sufficiently supported in the manuscript in terms of upscaling or cost effectiveness.

Reply: For our reply addressing the cost-effectiveness, please see comment 15.

REVIEWERS' COMMENTS third round:

Reviewer #2 (Remarks to the Author):

The authors have done a good job of addressing previous comments. I would like to see the following minor issues addressed, but otherwise am satisfied that the manuscript is ready for publication.

I appreciate the inclusion of a stats table in this version which mostly seems to match the text now. There is mention of a negative binomial model being used if needed, so I assume it was indeed used in one or more cases, and therefore should be specified in the table.

I also appreciate that the cost analysis now acknowledges the expense of using scuba at the seagrass sites. The authors have still not acknowledged that permitting agencies at sites with higher proportions of fine sediments as in many temperate seagrass locations could take issue with suspension of sediments that would occur with mat burial. These agencies could deny or greatly delay a project and/or require silt fences to maintain light availability in nearby seagrass patches. This is both a permitting complication/time issue and a cost issue. Further, placement of any material, even biodegradable, qualifies as fill in some regions; in my experience it would likely be allowed but would need to be approved by five or more regulatory agencies tasked with avoiding fill, avoiding impacts to species of concern including migratory shorebirds, and maintaining water quality standards. At least a cursory acknowledgment of these issues should be made.

Response letter to NCOMMS-19-7936145B

Dear editor and reviewers,

We wish to thank the reviewer once more for the helpful comments on our manuscript.

Following the suggestions of reviewer 2, we have now acknowledged possible concerns regarding a large-scale application of our approach in vulnerable ecosystems, and updated the statistics table.

We hope that we addressed all comments and suggestions to your satisfaction.

Sincerely,

Tjisse van der Heide, on behalf of all authors

Reviewer #2 (Remarks to the Author):

The authors have done a good job of addressing previous comments. I would like to see the following minor issues addressed, but otherwise am satisfied that the manuscript is ready for publication.

We are happy that you support our manuscript for publication.

1) I appreciate the inclusion of a stats table in this version which mostly seems to match the text now. There is mention of a negative binomial model being used if needed, so I assume it was indeed used in one or more cases, and therefore should be specified in the table.

Reply: We now specify for which dataset we used the negative binomial model, and updated the table accordingly.

L449-451: Poisson models were checked for overdispersion, and if unsatisfactory, a negative binomial model was used (Sweden data).

2) I also appreciate that the cost analysis now acknowledges the expense of using scuba at the seagrass sites. The authors have still not acknowledged that permitting agencies at sites with higher proportions of fine sediments as in many temperate seagrass locations could take issue with suspension of sediments that would occur with mat burial. These agencies could deny or greatly delay a project and/or require silt fences to maintain light availability in nearby seagrass patches. This is both a permitting complication/time issue and a cost issue. Further, placement of any material, even biodegradable, qualifies as fill in some regions; in my experience it would likely be allowed but would need to be approved by five or more regulatory agencies tasked with avoiding fill, avoiding impacts to species of concern including migratory shorebirds, and maintaining water quality standards. At least a cursory acknowledgment of these issues should be made.

Reply: We agree that restoration projects could face challenges with permitting due to the potential negative impacts associated with installing the biodegradable structures. Therefore, we now acknowledge this in the discussion:

L317-321: Finally, large-scale application should also be carefully judged in ecosystems that are suitable from an environmental perspective, but considered vulnerable

regarding for instance water and sediment quality, or the intermediate-term fate of biodegradable material. In such cases, permitting and mitigation measures could result in a prolonged project duration and higher costs.